# DistDF: Time-Series Forecasting Needs Joint-Distribution Wasserstein Alignment

Hao Wang[1,2]    Licheng Pan[1]    Yuan Lu[1]    Zhixuan Chu[3]    Xiaoxi Li[1]    Shuting He[4]
Zhichao Chen[5]    Qingsong Wen[6]    Haoxuan Li[7,†]    Zhouchen Lin[5,8,†]

[1]Xiaohongshu Inc.    [2]College of Control Science and Technology, Zhejiang University
[3]College of Computer Science and Technology, Zhejiang University
[4]School of Computing and Artificial Intelligence, Shanghai University of Finance and Economics
[5]State Key Lab of General AI, School of Intelligence Science and Technology, Peking University
[6]Squirrel AI    [7]Center for Data Science, Peking University
[8]Institute for Artificial Intelligence, Peking University

## Abstract

Training time-series forecasting models requires aligning the conditional distribution of model forecasts with that of the label sequence. The standard direct forecast (DF) approach resorts to minimizing the conditional negative log-likelihood, typically estimated by the mean squared error. However, this estimation proves biased when the label sequence exhibits autocorrelation. In this paper, we propose DistDF, which achieves alignment by minimizing a distributional discrepancy between the conditional distributions of forecast and label sequences. Since such conditional discrepancies are difficult to estimate from finite time-series observations, we introduce a joint-distribution Wasserstein discrepancy for time-series forecasting, which provably upper bounds the conditional discrepancy of interest. The proposed discrepancy is tractable, differentiable, and readily compatible with gradient-based optimization. Extensive experiments show that DistDF improves diverse forecasting models and achieves leading performance. Code is available at https://github.com/Master-PLC/DistDF.

## 1 Introduction

Time-series forecasting, which entails predicting future values based on historical observations, plays a critical role in numerous applications (Huang et al., 2026; Hu et al., 2025a), such as website traffic prediction in e-commerce (Chen et al., 2023), trajectory forecasting in robotics (Fan et al., 2023), and inferential sensing in manufacturing (Chen et al., 2025). In the era of deep learning, the development of effective forecasting models hinges on two aspects (Wang et al., 2026c): *(1) How to design neural architectures that serve as forecasting models?* and *(2) How to design learning objectives that drive model training?* Both aspects are essential for achieving high forecasting performance.

The design of neural architectures has been extensively investigated in recent studies. A central challenge involves effectively capturing the autocorrelation structures inherent in the input sequences. To this end, a variety of neural architectures have been proposed (Wu et al., 2024; Wang et al., 2024c). Recent work has focused on comparing Transformer-based models—which leverage self-attention mechanisms to model autocorrelation and scale effectively (Nie et al., 2023; Liu et al., 2024b; Wang et al., 2024a)—with linear models, which use linear projections to model autocorrelation and often achieve competitive performance with reduced complexity (Yi et al., 2023; Zeng et al., 2023; Yue et al., 2025). These developments illustrate a rapidly evolving landscape in time-series forecasting.

In contrast, the design of learning objectives remains less explored (Hu et al., 2026; Qiu et al., 2025a; Kudrat et al., 2025). Current approaches typically define the learning objective as the conditional likelihood of the label sequence. In practice, this is often estimated as the mean squared error (MSE), which has been a standard objective for training forecasting models (Lin et al., 2025). However,

---

†Corresponding authors.

MSE neglects the autocorrelation structure of the label sequence, leading to biased likelihood estimation (Wang et al., 2025d). Some efforts attempt to eliminate this bias by transforming the label sequence into conditionally decorrelated components (Wang et al., 2025c;d). However, existing label transformation methods cannot reliably guarantee conditional decorrelation, and the bias therefore persists. *Therefore, likelihood-based methods are fundamentally limited by biased likelihood estimation that impedes model training.*

To bypass the limitations of widespread likelihood-based methods, we propose Distribution-aware Direct Forecast (DistDF), which trains forecasting models by minimizing the discrepancy between the conditional distributions of forecast and label sequences. Since directly estimating conditional discrepancies is intractable given finite time-series observations, we introduce the joint-distribution Wasserstein discrepancy for training forecasting models. It upper-bounds the conditional discrepancy of interest, enables differentiation, and can be estimated from finite time-series observations, making it well-suited for integration with gradient-based optimization of time-series forecasting models.

Our main contributions are summarized as follows:

- We demonstrate a fundamental limitation in prevailing likelihood-based learning objectives for time-series forecasting: biased likelihood estimation that hampers effective model training.

- We propose DistDF, a training framework that aligns the conditional distributions of forecasts and labels, with a newly proposed joint-distribution Wasserstein discrepancy, ensuring the alignment of conditional distributions and admitting tractable estimation from finite time-series observations.

- We perform comprehensive empirical evaluations to demonstrate the effectiveness of DistDF, which enhances the performance of state-of-the-art forecasting models across diverse datasets.

## 2 PRELIMINARIES

### 2.1 PROBLEM DEFINITION

In this paper, we focus on the multi-step time-series forecasting problem (Wu et al., 2026). In general, we adhere to standard notational conventions: uppercase bold letters (*e.g.*, $\mathbf{X}$) denote matrices, lowercase bold letters (*e.g.*, $\mathbf{x}$) denote vectors, and lowercase normal letters (*e.g.*, $x$) denote scalars. Since the autocorrelation property central to our analysis manifests independently within each variate, we adopt the univariate setting for problem formulation and analysis (Nie et al., 2023; Wang et al., 2025c), which generalizes naturally to the multivariate setting.

Suppose $\mathbf{s} = [s_1, \ldots, s_M] \in \mathbb{R}^M$ is a time-series of M chronological observations. At time step $n$, we define the history sequence of length H as $\mathbf{x} = [s_{n-H+1}, \ldots, s_n] \in \mathbb{R}^H$ and the corresponding label sequence of length T as $\mathbf{y} = [s_{n+1}, \ldots, s_{n+T}] \in \mathbb{R}^T$. The goal of time-series forecasting is to learn a model $g : \mathbb{R}^H \to \mathbb{R}^T$ such that the forecast sequence $\hat{\mathbf{y}} = g(\mathbf{x})$ closely approximates $\mathbf{y}$.

The development of forecasting models encompasses two principal aspects: (1) neural network architectures that effectively encode history sequences (Zeng et al., 2023; Liu et al., 2024b), and (2) learning objectives for training neural networks (Qiu et al., 2025a; Wang et al., 2025d). It is important to emphasize that this work focuses on the design of learning objectives rather than proposing novel architectures. Nevertheless, we provide a concise review of both aspects for contextual completeness.

### 2.2 NEURAL NETWORK ARCHITECTURES IN TIME-SERIES FORECASTING

The development of network architectures aims to model the autocorrelation pattern in history sequences to obtain informative representations (Wu et al., 2024; Qiu et al., 2025b; Huang et al., 2024; 2025). Representative architectures include recurrent neural networks (Gu et al., 2021), convolutional neural networks (Luo and Wang, 2024), graph neural networks (Huang et al., 2023; Li et al., 2026; Liu et al., 2026; Ma et al., 2025b), and Transformers (Qiu et al., 2025c; Ma et al., 2025a; Nie et al., 2023). A central theme in recent literature is the comparison of Transformer and non-Transformer architectures. Transformers (*e.g.*, PatchTST (Nie et al., 2023), SRSNet (Wu et al., 2025a)) demonstrate strong scalability on large datasets but often entail substantial computational cost; non-Transformer models (*e.g.*, TimeMixer (Wang et al., 2024b), FreTS (Yi et al., 2023)) offer greater computational efficiency but may be less scalable. Recent advances include hybrid architectures that

combine Transformer and non-Transformer components for their complementary strengths (Lin et al., 2024), as well as the integration of Fourier analysis for efficient learning (Piao et al., 2024).

## 2.3 LEARNING OBJECTIVES IN TIME-SERIES FORECASTING

One widespread learning objective for training time-series forecasting models is MSE, which measures the point-wise error between the forecast and label sequences (Lin et al., 2025; Hu et al., 2025c):

$$\mathcal{L}_{\mathrm{MSE}} = \left\| \mathbf{y}_{|\mathbf{x}} - \hat{\mathbf{y}}_{|\mathbf{x}} \right\|_2^2 = \sum_{t=1}^{\mathrm{T}} \left( y_{|\mathbf{x},t} - \hat{y}_{|\mathbf{x},t} \right)^2, \tag{1}$$

where $\mathbf{y}_{|\mathbf{x}}$ is the label sequence given history sequence $\mathbf{x}$, $\hat{\mathbf{y}}_{|\mathbf{x}}$ is the forecast sequence. However, the MSE objective proves biased since it overlooks the presence of label autocorrelation (Wang et al., 2025d; 2026b). To mitigate this bias, alternative learning objectives are proposed. One line of work aligns the overall shape of the forecast and label sequence (*e.g.*, Dilate (Le Guen and Thome, 2019) and PS (Kudrat et al., 2025)). However, these methods often lack theoretical guarantees for achieving an unbiased objective. Another line of work transforms labels into decorrelated components before computing point-wise error. These methods reduce bias and improve forecasting performance (Wang et al., 2025c;d), showing the benefits of refining learning objectives for time-series forecasting.

## 3 METHODOLOGY

### 3.1 MOTIVATION

Training time-series forecasting models requires aligning the conditional distribution of model-generated forecasts with that of the label sequence. To achieve this, the dominant approach minimizes the conditional negative log-likelihood of $\mathbf{y}$, typically estimated using MSE (Wu et al., 2025b; Hu et al., 2025b; Ma et al., 2026a;b). However, MSE treats future steps in $\mathbf{y}$ as independent, ignoring label autocorrelation where $y_t$ depends on $y_{<t}$. This oversight renders MSE biased from the true negative log-likelihood of $\mathbf{y}$—an issue we term autocorrelation bias, formalized in Theorem 3.1.

**Theorem 3.1** (Autocorrelation bias). *Suppose* $\mathbf{y}_{|\mathbf{x}} \in \mathbb{R}^{\mathrm{T}}$ *is the label sequence given* $\mathbf{x}$, $\hat{\mathbf{y}}_{|\mathbf{x}} \in \mathbb{R}^{\mathrm{T}}$ *is the forecast sequence,* $\mathbf{\Sigma}_{|\mathbf{x}} \in \mathbb{R}^{\mathrm{T} \times \mathrm{T}}$ *is the conditional covariance of* $\mathbf{y}_{|\mathbf{x}}$. *The bias of MSE from the negative log-likelihood of the label sequence given* $\mathbf{x}$ *is expressed as:*

$$\mathrm{Bias} = \left\| \mathbf{y}_{|\mathbf{x}} - \hat{\mathbf{y}}_{|\mathbf{x}} \right\|_{\mathbf{\Sigma}_{|\mathbf{x}}^{-1}}^2 - \left\| \mathbf{y}_{|\mathbf{x}} - \hat{\mathbf{y}}_{|\mathbf{x}} \right\|_2^2. \tag{2}$$

*where* $\|\mathbf{v}\|_{\mathbf{\Sigma}_{|\mathbf{x}}^{-1}}^2 = \mathbf{v}^\top \mathbf{\Sigma}_{|\mathbf{x}}^{-1} \mathbf{v}$. *It vanishes if the conditional covariance* $\mathbf{\Sigma}_{|\mathbf{x}}$ *is the identity matrix*[1].

Some might argue that the bias can be eliminated by first transforming the label sequence into conditionally decorrelated components and then applying MSE component-wise. For example, **FreDF** (Wang et al., 2025d) uses Fourier transform to obtain frequency components; **Time-o1** (Wang et al., 2025c) employs principal component analysis to obtain principal components. This strategy does eliminate the bias if the resulting components were truly conditionally decorrelated (see Theorem 3.1). However, one key distinction warrants emphasis: both Fourier and principal component transformations guarantee only *marginal decorrelation* of the obtained components (*i.e.*, diagonal $\mathbf{\Sigma}$), not the required *conditional decorrelation* (*i.e.*, diagonal $\mathbf{\Sigma}_{|\mathbf{x}}$)[2]; thus the bias persists. *Hence, likelihood-based methods struggle with biased likelihood estimation which hampers model training.*

**Case study.** We examine the Traffic dataset to demonstrate limitations of existing likelihood-based methods. As shown in Fig. 1(a), the conditional correlation matrix reveals substantial off-diagonal values—over 50.3% exceed 0.1—illustrating the presence of autocorrelation effects. In contrast, Fig. 1(b-c) presents the conditional correlations of the latent components extracted by FreDF and

---

[1]The pioneering work (Wang et al., 2025c) derives the bias from the marginal likelihood of $\mathbf{y}$ assuming it follows a Gaussian distribution. In contrast, this work clarifies that it is preferable to treat the conditional distribution $\mathbb{P}_{\mathbf{y}|\mathbf{x}}$ as Gaussian. Consequently, we derive the bias from the conditional log-likelihood of $\mathbf{y}$.

[2]According to Theorem 3.3 (Wang et al., 2025d) and Lemma 3.2 (Wang et al., 2025c), the components obtained by Fourier and principal component transformations are marginally decorrelated.

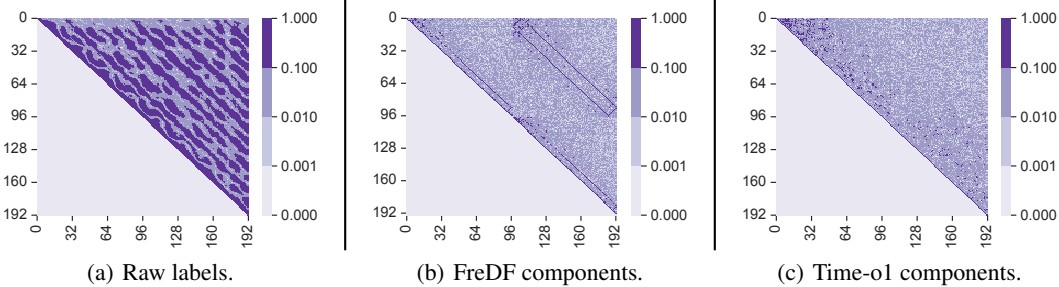

(a) Raw labels.  (b) FreDF components.  (c) Time-o1 components.

Figure 1: The conditional correlation of label components given $\mathbf{x}$, where the forecast length is set to $T = 192$. The correlation matrices are computed for the raw labels (a), the frequency components in FreDF (b) (Wang et al., 2025d) and the principal components in Time-o1 (c) (Wang et al., 2025c).

Time-o1 (Wang et al., 2025d;c). Although off-diagonal magnitudes are reduced, non-negligible residual correlations persist, implying incomplete removal of autocorrelation in the transformed components. Consequently, applying point-wise losses to these components still introduces bias.

Given the substantial challenges faced by likelihood-based methods, it is worthwhile to explore alternative strategies to align conditional distributions for training forecasting models. One plain strategy is directly minimizing a *distributional discrepancy between the conditional distributions* (Courty et al., 2017b), which can effectively achieve alignment while bypassing the complexity of likelihood estimation. This perspective raises two key questions warranting investigation: *How to devise a discrepancy to align two conditional distributions? Does it improve forecasting performance?*

## 3.2 ALIGNING CONDITIONAL DISTRIBUTIONS VIA JOINT-DISTRIBUTION BALANCING

In this section, we align the conditional distributions $\mathbb{P}_{\hat{\mathbf{y}}|\mathbf{x}}$ and $\mathbb{P}_{\mathbf{y}|\mathbf{x}}$ by minimizing a discrepancy metric between them. As in general distribution alignment tasks, the choice of discrepancy metric is crucial (Courty et al., 2017b). We select the Wasserstein discrepancy from optimal transport theory, which measures the minimum cost of transporting one distribution to another. Its rigorous theoretical properties and proven empirical success make it a principled choice for this work (Peyré and Cuturi, 2019). An informal definition is provided in Definition 3.2.

**Definition 3.2** (Wasserstein discrepancy). Let $\boldsymbol{\alpha}$ and $\boldsymbol{\beta}$ be random variables with probability distributions $\mathbb{P}_{\boldsymbol{\alpha}}$ and $\mathbb{P}_{\boldsymbol{\beta}}$, and let $\mathcal{S}_{\boldsymbol{\alpha}} = [\alpha_1, \ldots, \alpha_n]$ and $\mathcal{S}_{\boldsymbol{\beta}} = [\beta_1, \ldots, \beta_m]$ be empirical samples from $\mathbb{P}_{\boldsymbol{\alpha}}$ and $\mathbb{P}_{\boldsymbol{\beta}}$. The optimization problem below seeks a feasible plan $\mathbf{P} \in \mathbb{R}_+^{n \times m}$ to transport $\boldsymbol{\alpha}$ to $\boldsymbol{\beta}$ at the minimum cost:

$$\mathcal{W}_p(\mathbb{P}_{\boldsymbol{\alpha}}, \mathbb{P}_{\boldsymbol{\beta}}) := \min_{\mathbf{P} \in \Pi(\alpha, \beta)} \langle \mathbf{D}, \mathbf{P} \rangle,$$

$$\Pi(\mathbb{P}_{\boldsymbol{\alpha}}, \mathbb{P}_{\boldsymbol{\beta}}) := \begin{cases} p_{i,1} + \ldots + p_{i,m} = a_i, i = 1, \ldots, n, \\ p_{1,j} + \ldots + p_{n,j} = b_j, j = 1, \ldots, m, \\ p_{i,j} \geq 0, i = 1, \ldots, n, j = 1, \ldots, m, \end{cases} \tag{3}$$

*where $\mathcal{W}_p$ denotes the p-Wasserstein discrepancy; $\mathbf{D} \in \mathbb{R}_+^{n \times m}$ represents the pairwise distances calculated as $d_{i,j} = \|\alpha_i - \beta_j\|_p^p$; $\mathbf{a} = [a_1, \ldots, a_n]$ and $\mathbf{b} = [b_1, \ldots, b_m]$ are the weights of samples in $\boldsymbol{\alpha}$ and $\boldsymbol{\beta}$, respectively; n and m are the numbers of samples; $\Pi$ defines the set of constraints.*

A natural approach is to minimize $\mathcal{W}_p(\mathbb{P}_{\mathbf{y}|\mathbf{x}}, \mathbb{P}_{\hat{\mathbf{y}}|\mathbf{x}})$ to align the conditional distributions. However, it faces a fundamental *estimation difficulty*. For any given $\mathbf{x}$, a typical dataset often provides only a single associated label sequence $\mathbf{y}$, and the forecasting model produces only a single output $\hat{\mathbf{y}}$. Thus, the empirical sets ($\mathcal{S}_{\mathbf{y}|\mathbf{x}}$ and $\mathcal{S}_{\hat{\mathbf{y}}|\mathbf{x}}$) each contain only a single sample, which is insufficient to represent the underlying conditional distributions and renders the discrepancy uninformative.

**Lemma 3.3** (Kim et al. (2022)). *For any $p \geq 1$, the joint-distribution Wasserstein discrepancy upper bounds the expected conditional-distribution Wasserstein discrepancy:*

$$\int \mathcal{W}_p(\mathbb{P}_{\mathbf{y}|\mathbf{x}}, \mathbb{P}_{\hat{\mathbf{y}}|\mathbf{x}}) d\mathbb{P}(\mathbf{x}) \leq \mathcal{W}_p(\mathbb{P}_{\mathbf{x},\mathbf{y}}, \mathbb{P}_{\mathbf{x},\hat{\mathbf{y}}}). \tag{4}$$

*Equality holds if $p = 1$ or the conditional Wasserstein term is constant with respect to $\mathbf{x}$.*

---

**Algorithm 1** The workflow of DistDF.

---

**Input**: $\mathbf{X}$: a batch of history sequences, $\mathbf{Y}$: a batch of label sequences.
**Parameter**: $\gamma$: the relative weight of the discrepancy term, $g$: the forecasting model.
**Output**: $\mathcal{L}_{\mathrm{DistDF}}$: the obtained learning objective.

1: $\hat{\mathbf{Y}} \leftarrow g(\mathbf{X})$
2: $\mathbf{Z} \leftarrow \mathrm{concat}(\mathbf{X}, \mathbf{Y}), \hat{\mathbf{Z}} \leftarrow \mathrm{concat}(\mathbf{X}, \hat{\mathbf{Y}})$
3: $\boldsymbol{\mu}_{\mathbf{Z}} \leftarrow \mathrm{mean}(\mathbf{Z}), \boldsymbol{\Sigma}_{\mathbf{Z}} \leftarrow \mathrm{cov}(\mathbf{Z}), \boldsymbol{\mu}_{\hat{\mathbf{Z}}} \leftarrow \mathrm{mean}(\hat{\mathbf{Z}}), \boldsymbol{\Sigma}_{\hat{\mathbf{Z}}} \leftarrow \mathrm{cov}(\hat{\mathbf{Z}})$
4: $\mathcal{L}_{\mathrm{Dist}} \leftarrow \mathcal{BW}(\boldsymbol{\mu}_{\mathbf{Z}}, \boldsymbol{\mu}_{\hat{\mathbf{Z}}}, \boldsymbol{\Sigma}_{\mathbf{Z}}, \boldsymbol{\Sigma}_{\hat{\mathbf{Z}}})$
5: $\mathcal{L}_{\mathrm{MSE}} \leftarrow \|\mathbf{Y} - \hat{\mathbf{Y}}\|_2^2$
6: $\mathcal{L}_{\mathrm{DistDF}} := \gamma \cdot \mathcal{L}_{\mathrm{Dist}} + (1 - \gamma) \cdot \mathcal{L}_{\mathrm{MSE}}$

---

To bypass this estimation difficulty[3], we introduce the joint-distribution Wasserstein discrepancy for training time-series forecasting models (Courty et al., 2017a), denoted as $\mathcal{W}_p(\mathbb{P}_{\mathbf{x},\mathbf{y}}, \mathbb{P}_{\mathbf{x},\hat{\mathbf{y}}})$. This proxy offers two advantages. First, it provides a provable **upper bound** on the expected conditional discrepancy (see Lemma 3.3), ensuring that minimizing the joint discrepancy effectively promotes alignment of the conditional distributions of interest. Second, it is readily **estimable** from finite time-series observations: empirical sets $\mathcal{S}_{\mathbf{x},\mathbf{y}}$ and $\mathcal{S}_{\mathbf{x},\hat{\mathbf{y}}}$ can be constructed from the entire dataset, providing sufficient samples to compute an informative discrepancy.

**Theorem 3.4** (Alignment property). *The conditional distributions are aligned, i.e., $\mathbb{P}_{\mathbf{y}|\mathbf{x}} = \mathbb{P}_{\hat{\mathbf{y}}|\mathbf{x}}$ if the joint-distribution Wasserstein discrepancy is minimized to zero, i.e., $\mathcal{W}_p(\mathbb{P}_{\mathbf{x},\mathbf{y}}, \mathbb{P}_{\mathbf{x},\hat{\mathbf{y}}}) = 0$.*

**Lemma 3.5** (Peyré and Cuturi (2019)). *Suppose $\mathbb{P}_{\mathbf{x},\mathbf{y}}$ and $\mathbb{P}_{\mathbf{x},\hat{\mathbf{y}}}$ obey Gaussian distributions $\mathcal{N}(\boldsymbol{\mu}_{\mathbf{x},\mathbf{y}}, \boldsymbol{\Sigma}_{\mathbf{x},\mathbf{y}})$ and $\mathcal{N}(\boldsymbol{\mu}_{\mathbf{x},\hat{\mathbf{y}}}, \boldsymbol{\Sigma}_{\mathbf{x},\hat{\mathbf{y}}})$, respectively. The squared $\mathcal{W}_2$ discrepancy can be evaluated as the Bures–Wasserstein discrepancy:*

$$\mathcal{BW}(\boldsymbol{\mu}_{\mathbf{x},\mathbf{y}}, \boldsymbol{\mu}_{\mathbf{x},\hat{\mathbf{y}}}, \boldsymbol{\Sigma}_{\mathbf{x},\mathbf{y}}, \boldsymbol{\Sigma}_{\mathbf{x},\hat{\mathbf{y}}}) = \left\|\boldsymbol{\mu}_{\mathbf{x},\mathbf{y}} - \boldsymbol{\mu}_{\mathbf{x},\hat{\mathbf{y}}}\right\|_2^2 + \mathcal{B}(\boldsymbol{\Sigma}_{\mathbf{x},\mathbf{y}}, \boldsymbol{\Sigma}_{\mathbf{x},\hat{\mathbf{y}}}), \tag{5}$$

*where $\mathcal{B}(\boldsymbol{\Sigma}_{\mathbf{x},\mathbf{y}}, \boldsymbol{\Sigma}_{\mathbf{x},\hat{\mathbf{y}}}) = \mathrm{Tr}\left(\boldsymbol{\Sigma}_{\mathbf{x},\mathbf{y}} + \boldsymbol{\Sigma}_{\mathbf{x},\hat{\mathbf{y}}} - 2\sqrt{\boldsymbol{\Sigma}_{\mathbf{x},\mathbf{y}}^{1/2}\boldsymbol{\Sigma}_{\mathbf{x},\hat{\mathbf{y}}}\boldsymbol{\Sigma}_{\mathbf{x},\mathbf{y}}^{1/2}}\right)$, $\mathrm{Tr}(\cdot)$ denotes matrix trace.*

The use of Wasserstein discrepancy for distribution alignment is inspired by the domain adaptation literature (Courty et al., 2017b). However, one key distinction warrants emphasis. Domain adaptation primarily aligns the *marginal distributions of inputs* to improve cross-domain generalization; in contrast, we align the *conditional distributions* of model outputs and labels to perform multi-task learning, which represents a technically innovative strategy in multi-task supervised learning.

### 3.3 Model implementation

In this section, we implement DistDF, a framework for training time-series forecasting models by minimizing the joint-distribution Wasserstein discrepancy. The key steps are summarized in Algorithm 1.

Given a batch of history sequences $\mathbf{X} \in \mathbb{R}^{\mathrm{B}\times\mathrm{H}}$ and corresponding label sequences $\mathbf{Y} \in \mathbb{R}^{\mathrm{B}\times\mathrm{T}}$, where B denotes batch size, the forecasting model $g$ is employed to produce forecast sequences $\hat{\mathbf{Y}}$ (step 1). We then construct two joint sequences by concatenating $\mathbf{X}$ with $\mathbf{Y}$ and $\hat{\mathbf{Y}}$ along the time axis (step 2): $\mathbf{Z} = [\mathbf{X}, \mathbf{Y}]$ and $\hat{\mathbf{Z}} = [\mathbf{X}, \hat{\mathbf{Y}}]$. Next, we compute their first- and second-order statistics (step 3), namely the mean vectors ($\boldsymbol{\mu}_{\mathbf{Z}}, \boldsymbol{\mu}_{\hat{\mathbf{Z}}}$) and covariance matrices ($\boldsymbol{\Sigma}_{\mathbf{Z}}, \boldsymbol{\Sigma}_{\hat{\mathbf{Z}}}$). The discrepancy $\mathcal{L}_{\mathrm{Dist}}$ is then evaluated via the Bures–Wasserstein metric defined in Lemma 3.5 (step 4).

The moment-based discrepancy $\mathcal{L}_{\mathrm{Dist}}$ discards samplewise correspondences between $\mathbf{X}$ and $\mathbf{Y}$—information critical for training forecasting models. Consequently, DistDF incorporates $\mathcal{L}_{\mathrm{Dist}}$ as a regularization term alongside MSE following prior works (Wang et al., 2025c;d):

$$\mathcal{L}_{\mathrm{DistDF}} := \gamma \cdot \mathcal{L}_{\mathrm{Dist}} + (1 - \gamma) \cdot \mathcal{L}_{\mathrm{MSE}}, \tag{6}$$

where MSE preserves elementwise correspondences, $0 \leq \gamma \leq 1$ controls the weight of $\mathcal{L}_{\mathrm{Dist}}$.

---

[3]This limitation is not unique to the Wasserstein discrepancy; any distributional discrepancy metric becomes degenerate in the absence of multiple samples.

By minimizing the distributional discrepancy, DistDF aligns the conditional distributions of the forecast and label sequences, thereby effectively training the forecasting model. DistDF preserves the principal benefits of the canonical DF framework (Zeng et al., 2023; Liu et al., 2024b), such as efficient inference and multi-task learning capability. Moreover, DistDF is model-agnostic, which renders it a plug-and-play component to improve the training of different forecasting models.

## 4 EXPERIMENTS

To demonstrate the efficacy of DistDF, the following aspects deserve empirical investigation:

1. **Performance:** *Does DistDF perform well?* In Section 4.2, we benchmark DistDF against alternative learning objectives developed for training time-series forecasting models.

2. **Gain:** *Why does it work?* In Section 4.3, we perform an ablative study, dissecting the individual components of DistDF and clarifying their contributions to forecasting performance.

3. **Generality:** *Does it support other models and discrepancy measures?* In Section 4.4, we examine its compatibility with various models and discrepancies, with further results in Appendix D.4.

4. **Sensitivity:** *Is it sensitive to hyperparameters?* In Section 4.5, we analyze the sensitivity of DistDF to the hyperparameter $\gamma$, showing stable performance across a broad parameter range.

5. **Efficiency:** *What is its computational cost?* In Appendix D.5, we evaluate the running cost of DistDF across different scenarios.

### 4.1 SETUP

**Datasets.** We evaluate on standard datasets for time-series forecasting, including ETT, ECL, and Weather, chronologically split into training, validation, and test sets (Wang et al., 2025d;c). These datasets vary in dimensionality and temporal resolution and present unique challenges for forecasting. We fix the history length to 96 and consider four forecast lengths (T): 96, 192, 336, and 720. For final evaluation on the test set, the drop-last trick is disabled consistent with Qiu et al. (2024).

**Baselines.** As our focus is on the design of learning objectives, we consider competitive objectives for training forecasting models: (1) shape-alignment objectives, including DILATE (Le Guen and Thome, 2019) and Soft-DTW (Cuturi and Blondel, 2017); and (2) likelihood-based objectives, including Time-o1 (Wang et al., 2025c), Koopman (Lange et al., 2021), FreDF (Wang et al., 2025d), and DF (MSE). Implementations follow the associated official codebases.

**Implementation.** All models are trained with the Adam optimizer (Kingma and Ba, 2015), with early stopping applied if the validation loss does not improve for three consecutive epochs. When integrating DistDF into different forecasting models, we retain benchmark hyperparameters (Liu et al., 2024b; Piao et al., 2024), tuning only $\gamma \in (0, 1]$ and learning rate $\eta \in [5 \times 10^{-5}, 10^{-3}]$. Notably, adjusting the learning rate is necessary to account for dataset-dependent variations in the scale and gradient dynamics of the discrepancy term. Experiments are conducted on Intel Xeon Platinum 8383C CPUs with NVIDIA RTX H100 GPUs.

### 4.2 OVERALL PERFORMANCE

In this section, we compare DistDF with several established time-series learning objectives, using two forecasting models TimeBridge (Liu et al., 2025) and Fredformer (Piao et al., 2024) as testbeds. The results are presented in Table 1 with key observations as follows.

• The naive MSE objective exhibits suboptimal performance. For Fredformer, MSE (naive DF) yields the worst forecasting performance on ECL and Weather datasets. This stems from its neglect of label autocorrelation, rendering it a biased learning objective as discussed in Theorem 3.1.

• The shape-alignment objectives offer limited improvements over MSE. For example, for Time-Bridge, Soft-DTW outperforms DF on ECL but underperforms on other datasets. These methods

Table 1: Comparative results with other objectives for time-series forecasting.

| Loss | | **DistDF** | | Time-o1 | | FreDF | | Koopman | | Dilate | | Soft-DTW | | DF | |
|---|---|---|---|---|---|---|---|---|---|---|---|---|---|---|---|
| Metrics | | MSE | MAE | MSE | MAE | MSE | MAE | MSE | MAE | MSE | MAE | MSE | MAE | MSE | MAE |
| *TimeBridge* | ETTm1 | **0.383** | **0.397** | 0.383 | 0.397 | 0.386 | 0.398 | 0.460 | 0.438 | 0.387 | 0.400 | 0.395 | 0.402 | 0.387 | 0.400 |
| | ETTh1 | **0.434** | **0.436** | 0.439 | 0.438 | 0.439 | 0.436 | 0.459 | 0.449 | 0.464 | 0.452 | 0.452 | 0.445 | 0.442 | 0.440 |
| | ECL | **0.172** | **0.267** | 0.175 | 0.268 | 0.175 | 0.267 | 0.182 | 0.277 | 0.176 | 0.271 | 0.173 | 0.268 | 0.176 | 0.271 |
| | Weather | **0.248** | **0.275** | 0.250 | 0.275 | 0.254 | 0.276 | 0.269 | 0.293 | 0.252 | 0.277 | 0.260 | 0.280 | 0.252 | 0.277 |
| *Fredformer* | ETTm1 | **0.378** | 0.394 | 0.379 | **0.393** | 0.384 | 0.394 | 0.389 | 0.400 | 0.389 | 0.400 | 0.397 | 0.402 | 0.387 | 0.398 |
| | ETTh1 | **0.430** | **0.429** | 0.431 | 0.429 | 0.438 | 0.434 | 0.452 | 0.443 | 0.453 | 0.442 | 0.460 | 0.449 | 0.447 | 0.434 |
| | ECL | **0.173** | **0.266** | 0.178 | 0.270 | 0.179 | 0.272 | 0.190 | 0.282 | 0.187 | 0.280 | 0.206 | 0.298 | 0.191 | 0.284 |
| | Weather | **0.255** | 0.277 | 0.255 | **0.276** | 0.256 | 0.277 | 0.257 | 0.279 | 0.258 | 0.280 | 0.261 | 0.280 | 0.261 | 0.282 |

*Note*: **Bold** and underlined denote best and second-best results, respectively. The reported results are averaged over forecast lengths: T=96, 192, 336 and 720. When metric values coincide up to three decimal places, **Bold** indicates the numerically superior result based on full precision.

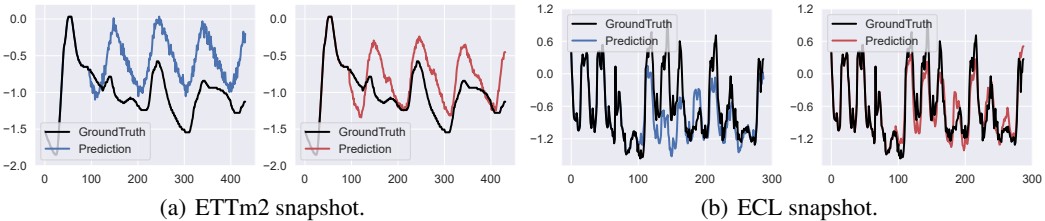

(a) ETTm2 snapshot.  (b) ECL snapshot.

Figure 2: The forecast sequence of DF (in blue) and DistDF (in red), with history length H = 96.

rely on heuristic geometric matching, which partially accommodates the autocorrelation structure of label sequences but fails to guarantee unbiasedness as learning objectives for conditional distribution alignment (Wang et al., 2026c), leading to suboptimal performance[4].

- The likelihood-based objectives offer more significant improvements over MSE. For example, FreDF and Time-o1 exhibit the best performance among baselines. This effectiveness is attributed to their reduction of autocorrelation bias relative to MSE. However, as discussed in Section 3.1, they fail to fully eliminate the bias, preventing true conditional distribution alignment.

- DistDF achieves the best overall performance among all compared objectives. By Theorem 3.4, it ensures unbiased alignment of conditional distributions without suffering from autocorrelation bias. This unique theoretical advantage translates into superior empirical performance across both forecasting models, yielding the lowest MSE in all investigated cases.

**Showcases.** To further illustrate the practical benefits, we perform a qualitative comparison of DF and DistDF in Fig. 2. While a model trained with the standard MSE objective captures the overall trend, it fails to accurately track fine-grained variations, such as rapid changes between steps 100 and 200. In contrast, DistDF produces forecasts that more precisely reflect these subtle and rapid changes, highlighting its effectiveness in improving real-world forecasting accuracy.

## 4.3 ABLATION STUDIES

In this section, we examine two components of the Wasserstein discrepancy in (5): mean alignment and covariance alignment. The results are presented in Table 2 with key observations as follows:

- Mean alignment improves conditional distribution matching. DistDF[†] augments DF by aligning the means of joint distributions via the mean difference term in (5). It outperforms DF, indicating that mean alignment facilitates conditional distribution alignment between label and forecast sequences.

- Covariance alignment also enhances conditional distribution alignment. DistDF[‡] enhances DF by aligning the variance of joint distributions via the $\mathcal{B}(\cdot)$ term in (5). It also yields improvements over DF in most cases, suggesting that variance alignment facilitates conditional distribution alignment.

---

[4]These observations align with those of Le Guen and Thome (2019).

Table 2: Ablation study results.

| Model | Align $\mu$ | Align $\Sigma$ | Data | T=96 | | T=192 | | T=336 | | T=720 | | Avg | |
|---|---|---|---|---|---|---|---|---|---|---|---|---|---|
| | | | | MSE | MAE | MSE | MAE | MSE | MAE | MSE | MAE | MSE | MAE |
| DF | ✗ | ✗ | ETTm1 | 0.326 | 0.361 | 0.365 | 0.382 | 0.396 | 0.404 | 0.459 | 0.444 | 0.387 | 0.398 |
| | | | ETTh1 | 0.377 | 0.396 | 0.437 | 0.425 | 0.486 | 0.449 | 0.447 | 0.467 | 0.447 | 0.434 |
| | | | ECL | 0.142 | 0.239 | 0.161 | 0.257 | 0.182 | 0.278 | 0.217 | 0.309 | 0.176 | 0.271 |
| | | | Weather | 0.168 | 0.211 | 0.214 | 0.254 | 0.273 | 0.297 | 0.353 | 0.347 | 0.252 | 0.277 |
| DistDF† | ✓ | ✗ | ETTm1 | 0.318 | 0.359 | 0.361 | 0.382 | 0.393 | 0.404 | 0.453 | 0.440 | 0.381 | 0.396 |
| | | | ETTh1 | 0.375 | 0.394 | 0.435 | 0.426 | 0.471 | 0.446 | 0.457 | 0.455 | 0.435 | 0.430 |
| | | | ECL | 0.142 | 0.239 | 0.160 | 0.257 | 0.180 | 0.273 | 0.217 | 0.307 | 0.175 | 0.269 |
| | | | Weather | 0.168 | 0.211 | 0.213 | 0.253 | 0.273 | 0.296 | 0.349 | 0.348 | 0.251 | 0.277 |
| DistDF‡ | ✗ | ✓ | ETTm1 | 0.328 | 0.365 | 0.364 | 0.385 | 0.395 | 0.406 | 0.457 | 0.441 | 0.386 | 0.399 |
| | | | ETTh1 | 0.374 | 0.396 | 0.430 | 0.430 | 0.476 | 0.451 | 0.476 | 0.472 | 0.439 | 0.437 |
| | | | ECL | 0.141 | 0.239 | 0.161 | 0.257 | 0.179 | 0.273 | 0.216 | 0.307 | 0.174 | 0.269 |
| | | | Weather | 0.168 | 0.211 | 0.214 | 0.253 | 0.270 | 0.296 | 0.353 | 0.347 | 0.251 | 0.277 |
| DistDF | ✓ | ✓ | ETTm1 | **0.316** | **0.357** | **0.359** | **0.381** | **0.392** | **0.404** | **0.448** | **0.437** | **0.379** | **0.395** |
| | | | ETTh1 | **0.373** | **0.393** | **0.428** | **0.425** | **0.466** | **0.445** | **0.453** | **0.453** | **0.430** | **0.429** |
| | | | ECL | **0.137** | **0.235** | **0.159** | **0.257** | **0.178** | **0.272** | **0.212** | **0.302** | **0.172** | **0.267** |
| | | | Weather | **0.164** | **0.209** | **0.212** | **0.252** | **0.270** | **0.295** | **0.348** | **0.345** | **0.248** | **0.275** |

*Note*: **Bold** and underlined denote best and second-best results, respectively. When metric values coincide up to three decimal places, **Bold** indicates the numerically superior result based on full precision.

Table 3: Comparative results with other discrepancies for aligning the joint distributions.

| Discrepancy | | **Ours** | | EMD | | MMD@Linear | | MMD@RBF | | KL | | DF | |
|---|---|---|---|---|---|---|---|---|---|---|---|---|---|
| Metrics | | MSE | MAE | MSE | MAE | MSE | MAE | MSE | MAE | MSE | MAE | MSE | MAE |
| TimeBridge | ETTm1 | **0.383** | **0.398** | 0.388 | 0.400 | 0.385 | 0.400 | 0.387 | 0.399 | 0.387 | 0.400 | 0.387 | 0.400 |
| | ETTh1 | **0.433** | 0.437 | 0.441 | 0.439 | 0.438 | **0.437** | 0.441 | 0.440 | 0.437 | 0.438 | 0.442 | 0.440 |
| | ECL | **0.172** | 0.267 | 0.177 | 0.272 | 0.174 | 0.269 | 0.172 | **0.266** | 0.176 | 0.271 | 0.176 | 0.271 |
| | Weather | **0.248** | **0.275** | 0.251 | 0.276 | 0.253 | 0.278 | 0.250 | 0.276 | 0.253 | 0.277 | 0.252 | 0.277 |
| Fredformer | ETTm1 | **0.379** | **0.395** | 0.386 | 0.397 | 0.380 | 0.395 | 0.385 | 0.397 | 0.385 | 0.397 | 0.387 | 0.398 |
| | ETTh1 | **0.429** | **0.431** | 0.445 | 0.435 | 0.437 | 0.432 | 0.444 | 0.435 | 0.444 | 0.435 | 0.447 | 0.434 |
| | ECL | **0.183** | **0.275** | 0.187 | 0.280 | 0.188 | 0.280 | 0.187 | 0.280 | 0.187 | 0.279 | 0.191 | 0.284 |
| | Weather | **0.257** | **0.279** | 0.261 | 0.282 | 0.262 | 0.282 | 0.262 | 0.282 | 0.261 | 0.282 | 0.261 | 0.282 |

*Note*: **Bold** and underlined denote best and second-best results, respectively. The reported results are averaged over forecast lengths: T=96, 192, 336 and 720. When metric values coincide up to three decimal places, **Bold** indicates the numerically superior result based on full precision. Abbreviations: KL refers to Kullback–Leibler divergence; MMD refers to maximum mean discrepancy with different kernels; EMD refers to the earth mover's distance.

- Combining both components achieves synergistic gains. DistDF integrates both mean and covariance alignment for comprehensive joint distribution matching. It achieves the best performance, demonstrating a synergistic effect when both components are combined for alignment.

## 4.4 GENERALIZATION STUDIES

In this section, we assess the generalizability of DistDF by implementing it with different distribution discrepancy measures and integrating it with different forecasting models.

- Firstly, we evaluate DistDF using various discrepancy measures for joint distribution alignment. As shown in Table 3, all discrepancy measures yield improvements over MSE, confirming the benefit of incorporating distribution alignment in training forecasting models. The joint-distribution Wasserstein discrepancy achieves the best performance in 14 out of 16 cases, highlighting its effectiveness and reliability in distribution alignment (Peyré and Cuturi, 2019).

- Secondly, we evaluate DistDF when integrated to train different forecasting models. As shown in Fig. 3, DistDF consistently enhances forecasting performance across all forecasting models. For example, on the ECL dataset, iTransformer and Fredformer achieve substantial MSE reductions of up to 2.7% and 4.3%, respectively, when augmented with DistDF. These results underscore DistDF's potential as a general, plug-and-play enhancement for a wide range of forecasting models.

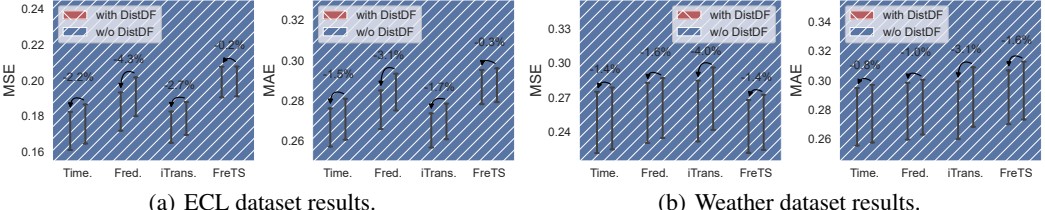

(a) ECL dataset results.  (b) Weather dataset results.

Figure 3: Improvement of DistDF applied to different forecasting models, shown with colored bars for means over forecast lengths (96, 192, 336, 720) and error bars for 50% confidence intervals.

Table 4: Varying $\gamma$ results of TimeBridge

| $\gamma$ | ETTh2 | | ECL | | Weather | |
|---|---|---|---|---|---|---|
| | MSE | MAE | MSE | MAE | MSE | MAE |
| 0 | 0.377 | 0.403 | 0.176 | 0.271 | 0.252 | 0.277 |
| 0.001 | 0.378 | 0.402 | 0.172 | 0.267 | 0.250 | 0.276 |
| 0.002 | 0.377 | 0.402 | 0.173 | 0.267 | 0.250 | 0.276 |
| 0.005 | 0.376 | 0.401 | **0.172** | **0.267** | 0.250 | **0.276** |
| 0.01 | 0.376 | 0.400 | 0.172 | 0.267 | **0.249** | 0.276 |
| 0.02 | 0.376 | 0.400 | 0.174 | 0.269 | 0.249 | 0.276 |
| 0.05 | **0.375** | 0.399 | 0.174 | 0.268 | 0.252 | 0.278 |
| 0.1 | 0.375 | **0.399** | 0.174 | 0.269 | 0.254 | 0.280 |
| 0.2 | 0.376 | 0.399 | 0.177 | 0.270 | 0.258 | 0.282 |
| 0.5 | 0.378 | 0.400 | 0.186 | 0.277 | 0.261 | 0.285 |
| 1 | 0.381 | 0.402 | 0.197 | 0.282 | 0.265 | 0.286 |

*Note*: **Bold** and underlined denote the best and second-best results.

Table 5: Varying $\gamma$ results of Fredformer.

| $\gamma$ | ETTh2 | | ECL | | Weather | |
|---|---|---|---|---|---|---|
| | MSE | MAE | MSE | MAE | MSE | MAE |
| 0 | 0.377 | 0.402 | 0.191 | 0.284 | 0.261 | 0.282 |
| 0.001 | 0.371 | 0.397 | 0.182 | 0.275 | 0.257 | 0.279 |
| 0.002 | 0.372 | 0.398 | **0.181** | **0.274** | 0.257 | 0.279 |
| 0.005 | 0.372 | 0.398 | 0.182 | 0.275 | 0.257 | 0.280 |
| 0.01 | 0.370 | 0.397 | 0.183 | 0.275 | **0.257** | **0.279** |
| 0.02 | **0.369** | **0.395** | 0.182 | 0.275 | 0.258 | 0.280 |
| 0.05 | 0.370 | 0.396 | 0.187 | 0.279 | 0.259 | 0.281 |
| 0.1 | 0.371 | 0.397 | 0.196 | 0.287 | 0.261 | 0.283 |
| 0.2 | 0.372 | 0.398 | 0.209 | 0.298 | 0.263 | 0.285 |
| 0.5 | 0.376 | 0.399 | 0.230 | 0.317 | 0.266 | 0.287 |
| 1 | 0.386 | 0.406 | 0.239 | 0.326 | 0.268 | 0.290 |

*Note*: **Bold** and underlined denote the best and second-best results.

## 4.5 HYPERPARAMETER SENSITIVITY

In this section, we analyze the impact of the distribution alignment weight $\gamma$ on DistDF's performance. As shown in Table 4 and Table 5, increasing $\gamma$ from zero generally yields improved performance. For instance, Fredformer achieves a 0.01 reduction in MSE on the ECL dataset, demonstrating the benefit of incorporating the discrepancy term into the learning objective. Moreover, the optimal performance is typically observed when $\gamma < 1$, highlighting the complementary role of the MSE term, which is easy to optimize and effective in ensuring point-wise alignment between forecast and label sequences.

## CONCLUSION

In this study, we demonstrate that existing likelihood-based approaches suffer from biased likelihood estimation. Instead, we propose DistDF, which trains forecasting models by minimizing the discrepancy between the conditional distributions of forecasts and labels. Recognizing the intractability of directly estimating conditional discrepancies, we introduce a joint-distribution Wasserstein discrepancy that serves as a tractable upper bound and can be efficiently estimated from observed data. Minimizing this quantity provably aligns conditional distributions for training forecasting models. Extensive experiments show that DistDF consistently improves forecasting performance.

**Limitations.** According to Lemma 3.5, DistDF measures distributional discrepancy via the Bures–Wasserstein discrepancy under a Gaussian assumption, yielding a tractable formulation. However, this metric captures only first- and second-order moments (mean and covariance). While sufficient for Gaussian distributions, real-world data may exhibit non-Gaussian characteristics that require higher-order moments for full characterization. Nevertheless, mean and covariance remain fundamental descriptors, and the Bures–Wasserstein discrepancy can still provide effective alignment by matching these moments. Extending DistDF to incorporate discrepancies that capture higher-order statistics while remaining computationally tractable is a promising direction for future work.

ACKNOWLEDGMENTS

Z. Lin was supported by the NSF China (No. 62276004), the Beijing Natural Science Foundation (No. L257007), and the Beijing Major Science and Technology Project (No. Z251100008425006).

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

## A    THEORETICAL JUSTIFICATION

**Theorem A.1** (Autocorrelation bias). *Suppose $\mathbf{y}_{|\mathbf{x}} \in \mathbb{R}^{\mathrm{T}}$ is the label sequence given $\mathbf{x}$, $\hat{\mathbf{y}}_{|\mathbf{x}} \in \mathbb{R}^{\mathrm{T}}$ is the forecast sequence, $\boldsymbol{\Sigma}_{|\mathbf{x}} \in \mathbb{R}^{\mathrm{T}\times\mathrm{T}}$ is the conditional covariance of $\mathbf{y}_{|\mathbf{x}}$. The bias of MSE from the negative log-likelihood of the label sequence given $\mathbf{x}$ is expressed as:*

$$\text{Bias} = \left\| \mathbf{y}_{|\mathbf{x}} - \hat{\mathbf{y}}_{|\mathbf{x}} \right\|_{\boldsymbol{\Sigma}_{|\mathbf{x}}^{-1}}^{2} - \left\| \mathbf{y}_{|\mathbf{x}} - \hat{\mathbf{y}}_{|\mathbf{x}} \right\|_{2}^{2}. \tag{7}$$

*where $\|\mathbf{v}\|_{\boldsymbol{\Sigma}_{|\mathbf{x}}^{-1}}^{2} = \mathbf{v}^{\top} \boldsymbol{\Sigma}_{|\mathbf{x}}^{-1} \mathbf{v}$. It vanishes if the conditional covariance $\boldsymbol{\Sigma}_{|\mathbf{x}}$ is the identity matrix.*

*Proof.* The proof follows our previous work (Wang et al., 2025c) but highlights that it is the conditional distribution of $\mathbf{y}$ given $\mathbf{x}$ that obeys Gaussian distribution, instead of the marginal distribution of $\mathbf{y}$. Suppose the label sequence given $\mathbf{x}$ follows a multivariate normal distribution with mean vector $\hat{\mathbf{y}}_{|\mathbf{x}} \in \mathbb{R}^{\mathrm{T}}$ and covariance matrix $\boldsymbol{\Sigma}_{|\mathbf{x}} \in \mathbb{R}^{\mathrm{T}\times\mathrm{T}}$. The conditional likelihood of $\mathbf{y}$ is:

$$\mathbb{P}_{\mathbf{y}|\mathbf{x}} = \frac{1}{(2\pi)^{0.5\mathrm{T}} |\boldsymbol{\Sigma}_{|\mathbf{x}}|^{0.5}} \exp(-\frac{1}{2} \left\| \mathbf{y}_{|\mathbf{x}} - \hat{\mathbf{y}}_{|\mathbf{x}} \right\|_{\boldsymbol{\Sigma}_{|\mathbf{x}}^{-1}}^{2}) \tag{8}$$

On the basis, the conditional negative log-likelihood of $\mathbf{y}$ is:

$$- \log \mathbb{P}_{\mathbf{y}|\mathbf{x}} = \frac{1}{2} \left( \mathrm{T}\log(2\pi) + \log |\boldsymbol{\Sigma}_{|\mathbf{x}}| + \left\| \mathbf{y}_{|\mathbf{x}} - \hat{\mathbf{y}}_{|\mathbf{x}} \right\|_{\boldsymbol{\Sigma}_{|\mathbf{x}}^{-1}}^{2} \right).$$

Removing the terms unrelated to $\hat{\mathbf{y}}_{|\mathbf{x}}$, the terms used for updating $\hat{\mathbf{y}}_{|\mathbf{x}}$, namely practical negative log-likelihood (PNLL), is expressed as follows:

$$\text{PNLL} = \left\| \mathbf{y}_{|\mathbf{x}} - \hat{\mathbf{y}}_{|\mathbf{x}} \right\|_{\boldsymbol{\Sigma}_{|\mathbf{x}}^{-1}}^{2}. \tag{9}$$

On the other hand, the MSE loss can be expressed as:

$$\text{MSE} = \left\| \mathbf{y}_{|\mathbf{x}} - \hat{\mathbf{y}}_{|\mathbf{x}} \right\|_{2}^{2}. \tag{10}$$

The difference between PNLL and MSE is computed as:

$$\text{Bias} = \left\| \mathbf{y}_{|\mathbf{x}} - \hat{\mathbf{y}}_{|\mathbf{x}} \right\|_{\boldsymbol{\Sigma}_{|\mathbf{x}}^{-1}}^{2} - \left\| \mathbf{y}_{|\mathbf{x}} - \hat{\mathbf{y}}_{|\mathbf{x}} \right\|^{2}, \tag{11}$$

which diminishes to zero if the label sequence is conditionally decorrelated, *i.e.*, $\boldsymbol{\Sigma}_{|\mathbf{x}}$ is identity matrix. The proof is completed. $\square$

**Lemma A.2** (Lemma 3.3 in the main text). *For any $p \geq 1$, the joint-distribution Wasserstein discrepancy upper bounds the expected conditional-distribution Wasserstein discrepancy:*

$$\int \mathcal{W}_p(\mathbb{P}_{\mathbf{y}|\mathbf{x}}, \mathbb{P}_{\hat{\mathbf{y}}|\mathbf{x}}) d\mathbb{P}(\mathbf{x}) \leq \mathcal{W}_p(\mathbb{P}_{\mathbf{x},\mathbf{y}}, \mathbb{P}_{\mathbf{x},\hat{\mathbf{y}}}). \tag{12}$$

*where the equality holds if $p = 1$ or the conditional Wasserstein term is constant with respect to $\mathbf{x}$.*

*Proof.* The proof can be found in Theorem 2 of Kim et al. (2022). $\square$

**Theorem A.3** (Alignment property). *The conditional distributions are aligned, i.e., $\mathbb{P}_{\mathbf{y}|\mathbf{x}} = \mathbb{P}_{\hat{\mathbf{y}}|\mathbf{x}}$ if the joint-distribution Wasserstein discrepancy is minimized to zero, i.e., $\mathcal{W}_p(\mathbb{P}_{\mathbf{x},\mathbf{y}}, \mathbb{P}_{\mathbf{x},\hat{\mathbf{y}}}) = 0$.*

*Proof.* By Lemma 3.3, we have

$$\int \mathcal{W}_p(\mathbb{P}_{\mathbf{y}|\mathbf{x}}, \mathbb{P}_{\hat{\mathbf{y}}|\mathbf{x}}) d\mathbb{P}(\mathbf{x}) \leq \mathcal{W}_p(\mathbb{P}_{\mathbf{x},\mathbf{y}}, \mathbb{P}_{\mathbf{x},\hat{\mathbf{y}}}).$$

Thus, if RHS $= 0$, we have $\int \mathcal{W}_p(\mathbb{P}_{\mathbf{y}|\mathbf{x}}, \mathbb{P}_{\hat{\mathbf{y}}|\mathbf{x}}) d\mathbb{P}(\mathbf{x}) = 0$. Since $\mathcal{W}_p$ is non-negative (Peyré and Cuturi, 2019), this implies that $\mathcal{W}_p(\mathbb{P}_{\mathbf{y}|\mathbf{x}}, \mathbb{P}_{\hat{\mathbf{y}}|\mathbf{x}}) = 0$ for almost every $\mathbf{x}$. Therefore, it suffices to prove that for two distributions $\mathbb{P}_{\boldsymbol{\alpha}} = \mathbb{P}_{\mathbf{y}|\mathbf{x}}$ and $\mathbb{P}_{\boldsymbol{\beta}} = \mathbb{P}_{\hat{\mathbf{y}}|\mathbf{x}}$, $\mathcal{W}_p(\mathbb{P}_{\boldsymbol{\alpha}}, \mathbb{P}_{\boldsymbol{\beta}}) = 0$ implies $\mathbb{P}_{\boldsymbol{\alpha}} = \mathbb{P}_{\boldsymbol{\beta}}$.

Suppose $\mathcal{S}_{\boldsymbol{\alpha}} = [\alpha_1, ..., \alpha_n]$ and $\mathcal{S}_{\boldsymbol{\beta}} = [\beta_1, ..., \beta_m]$ are the empirical samples from $\mathbb{P}_{\boldsymbol{\alpha}}$ and $\mathbb{P}_{\boldsymbol{\beta}}$, respectively, with corresponding mass vectors $\mathbf{a} \in \mathbb{R}^n$ and $\mathbf{b} \in \mathbb{R}^m$. We are given that $\mathcal{W}_p(\mathbb{P}_{\boldsymbol{\alpha}}, \mathbb{P}_{\boldsymbol{\beta}}) = 0$. By Definition 3.2, this means the minimum value of the cost function is zero. Let $\mathbf{P}^*$ be an optimal transport plan that solves the minimization problem; we have:

$$\mathcal{W}_p(\mathbb{P}_{\boldsymbol{\alpha}}, \mathbb{P}_{\boldsymbol{\beta}}) = \langle \mathbf{D}, \mathbf{P}^* \rangle = \sum_{i=1}^n \sum_{j=1}^m p_{i,j}^* \|\alpha_i - \beta_j\|_p^p = 0. \tag{13}$$

By the last constraint in (3), we have $p_{i,j}^* \geq 0$. The distance term is also non-negative: $\|\alpha_i - \beta_j\|_p^p \geq 0$. Since the sum of these non-negative terms is zero, each individual term below must be zero:

$$p_{i,j}^* \|\alpha_i - \beta_j\|_p^p = 0, \quad \forall i = 1, ..., n, j = 1, ..., m, \tag{14}$$

which implies that if any mass is transported from a point $\alpha_i$ to another point $\beta_j$, then the distance between the two points must be zero (*i.e.*, if $p_{i,j}^* > 0$, then $\alpha_i = \beta_j$). That is, the optimal plan only transports mass between identical points.

Consider an arbitrary value $z$ in the support of either distribution. The total mass at $z$, *i.e.*, probability density, for distribution $\mathbb{P}_{\boldsymbol{\alpha}}$ is $\mathbb{P}_{\boldsymbol{\alpha}}(z) = \sum_{i:\alpha_i=z} a_i$. By the constraints in (3), we can express this as:

$$\mathbb{P}_{\boldsymbol{\alpha}}(z) = \sum_{i:\alpha_i=z} a_i = \sum_{i:\alpha_i=z} \left( \sum_{j=1}^m p_{i,j}^* \right). \tag{15}$$

As established, $p_{i,j}^*$ can only be non-zero if $\beta_j = \alpha_i$. Therefore, for the outer sum where $\alpha_i = z$, the inner sum over $j$ is non-zero only for those indices $j$ where $\beta_j = z$. Thus, we can write:

$$\mathbb{P}_{\boldsymbol{\alpha}}(z) = \sum_{i:\alpha_i=z} \sum_{j:\beta_j=z} p_{i,j}^*. \tag{16}$$

Similarly, the mass at $z$ for distribution $\mathbb{P}_{\boldsymbol{\beta}}$ is $\mathbb{P}_{\boldsymbol{\beta}}(z) = \sum_{j:\beta_j=z} b_j$. By the constraints in (3):

$$\mathbb{P}_{\boldsymbol{\beta}}(z) = \sum_{j:\beta_j=z} b_j = \sum_{j:\beta_j=z} \left( \sum_{i=1}^n p_{i,j}^* \right) = \sum_{j:\beta_j=z} \sum_{i:\alpha_i=z} p_{i,j}^*, \tag{17}$$

which yields $\mathbb{P}_{\boldsymbol{\alpha}}(z) = \mathbb{P}_{\boldsymbol{\beta}}(z)$. Since this equality holds for any value $z$, the probability mass functions of $\mathbb{P}_{\boldsymbol{\alpha}}$ and $\mathbb{P}_{\boldsymbol{\beta}}$ are identical, which implies $\mathbb{P}_{\boldsymbol{\alpha}} = \mathbb{P}_{\boldsymbol{\beta}}$[5].

Applying this result to our conditional distributions, $\mathcal{W}_p(\mathbb{P}_{\mathbf{y}|\mathbf{x}}, \mathbb{P}_{\hat{\mathbf{y}}|\mathbf{x}}) = 0$ implies $\mathbb{P}_{\mathbf{y}|\mathbf{x}} = \mathbb{P}_{\hat{\mathbf{y}}|\mathbf{x}}$ for almost every $\mathbf{x}$. This completes the proof. $\square$

**Lemma A.4.** *Suppose* $\mathbb{P}_{\mathbf{x},\mathbf{y}}$ *and* $\mathbb{P}_{\mathbf{x},\hat{\mathbf{y}}}$ *obey Gaussian distributions* $\mathcal{N}(\boldsymbol{\mu}_{\mathbf{x},\mathbf{y}}, \boldsymbol{\Sigma}_{\mathbf{x},\mathbf{y}})$ *and* $\mathcal{N}(\boldsymbol{\mu}_{\mathbf{x},\hat{\mathbf{y}}}, \boldsymbol{\Sigma}_{\mathbf{x},\hat{\mathbf{y}}})$, *respectively. The squared* $\mathcal{W}_2$ *discrepancy can be calculated as the Bures-Wasserstein discrepancy:*

$$\mathcal{BW}(\boldsymbol{\mu}_{\mathbf{x},\mathbf{y}}, \boldsymbol{\mu}_{\mathbf{x},\hat{\mathbf{y}}}, \boldsymbol{\Sigma}_{\mathbf{x},\mathbf{y}}, \boldsymbol{\Sigma}_{\mathbf{x},\hat{\mathbf{y}}}) = \left\| \boldsymbol{\mu}_{\mathbf{x},\mathbf{y}} - \boldsymbol{\mu}_{\mathbf{x},\hat{\mathbf{y}}} \right\|_2^2 + \mathcal{B}(\boldsymbol{\Sigma}_{\mathbf{x},\mathbf{y}}, \boldsymbol{\Sigma}_{\mathbf{x},\hat{\mathbf{y}}}), \tag{18}$$

*where* $\mathcal{B}(\boldsymbol{\Sigma}_{\mathbf{x},\mathbf{y}}, \boldsymbol{\Sigma}_{\mathbf{x},\hat{\mathbf{y}}}) = \mathrm{Tr}\left( \boldsymbol{\Sigma}_{\mathbf{x},\mathbf{y}} + \boldsymbol{\Sigma}_{\mathbf{x},\hat{\mathbf{y}}} - 2\sqrt{\boldsymbol{\Sigma}_{\mathbf{x},\mathbf{y}}^{1/2} \boldsymbol{\Sigma}_{\mathbf{x},\hat{\mathbf{y}}} \boldsymbol{\Sigma}_{\mathbf{x},\mathbf{y}}^{1/2}} \right)$, $\mathrm{Tr}(\cdot)$ *denotes matrix trace.*

*Proof.* The proof can be found in Remark 2.31 of Peyré and Cuturi (2019). $\square$

# B DISCRETE OPTIMAL TRANSPORT AND WASSERSTEIN DISCREPANCY

In this section, we introduce the foundational concepts of optimal transport (OT) and the Wasserstein discrepancy. Our analysis is specifically confined to discrete probability measures, as the broader

---

[5]A discrete probability is completely characterized by two components: its support and its probability mass function.

theory involving general measures is beyond the scope of this work. For a comprehensive treatment of the continuous case, readers are directed to the seminal works by Peyré and Cuturi (2019).

The classical framing of OT, known as the Monge problem, can be illustrated with a simple scenario: transporting goods from n warehouses to m factories (Peyré and Cuturi, 2019). Let the $i$-th warehouse hold $a_i$ units of material and the $j$-th factory require $b_j$ units. The objective is to find a transport map that moves all material from the warehouses to satisfy the factories' demands. This problem is subject to several constraints: the entire stock from each warehouse must be shipped, all factory demands must be met, and the mapping must be deterministic (i.e., each warehouse ships its entire stock to a single factory). The optimal map is the one that minimizes the total cost, which is aggregated from the cost of moving a unit of material from a given warehouse to a factory.

**Definition B.1** (Monge Problem for Discrete Measures). Let $\boldsymbol{\alpha} = \sum_{i=1}^{n} a_i \delta_{\alpha_i}$ and $\boldsymbol{\beta} = \sum_{j=1}^{m} b_j \delta_{\beta_j}$ be two discrete probability measures. The Monge problem seeks a transport map $\mathbb{T}$ that pushes the mass of $\boldsymbol{\alpha}$ forward to match $\boldsymbol{\beta}$, denoted by $\mathbb{T}_\sharp \boldsymbol{\alpha} = \boldsymbol{\beta}$. This condition implies that for each $j$, the total mass received ($b_j$) must equal the sum of the masses sent from all locations mapped to it: $b_j = \sum_{i:\mathbb{T}(\alpha_i)=\beta_j} a_i$. The objective is to find the map $\mathbb{T}$ that minimizes the total transportation cost:

$$\min_{\mathbb{T}:\mathbb{T}_\sharp \boldsymbol{\alpha}=\boldsymbol{\beta}} \left\{ \sum_{i=1}^{n} d(\alpha_i, \mathbb{T}(\alpha_i)) a_i \right\}. \tag{19}$$

While intuitive, the Monge formulation is restrictive; a solution is not guaranteed to exist, particularly when mass splitting is required (e.g., one warehouse supplying multiple factories). To address this limitation, Kantorovich (2006) introduced a relaxed formulation. Instead of a deterministic map, Kantorovich's approach seeks a "transport plan" that allows mass from a single source to be distributed among multiple destinations. This recasts the problem as a linear programming problem.

**Definition B.2** (Kantorovich Problem). *Let $\boldsymbol{\alpha} = \sum_{i=1}^{n} a_i \delta_{\alpha_i}$ and $\boldsymbol{\beta} = \sum_{j=1}^{m} b_j \delta_{\beta_j}$ be two discrete probability distributions supported on samples $\{\alpha_i\}_{i=1}^{n}$ and $\{\beta_j\}_{j=1}^{m}$, respectively. The optimal transport problem is to find a transport plan $\mathbf{P} \in \mathbb{R}_+^{n \times m}$ that minimizes the total cost:*

$$\mathcal{W}_c(\boldsymbol{\alpha}, \boldsymbol{\beta}) := \min_{\mathbf{P} \in \Pi(\mathbf{a},\mathbf{b})} \langle \mathbf{D}, \mathbf{P} \rangle_F, \tag{20}$$

*where $\langle \cdot, \cdot \rangle_F$ is the Frobenius dot product. The cost matrix $\mathbf{D} \in \mathbb{R}_+^{n \times m}$ contains the pairwise costs, e.g., $d_{i,j} = c(\alpha_i, \beta_j)$. The set of feasible transport plans, $\Pi(\mathbf{a}, \mathbf{b})$, is defined by the constraints that preserve the total mass of the source and target measures:*

$$\Pi(\mathbf{a}, \mathbf{b}) := \left\{ \mathbf{P} \in \mathbb{R}_+^{n \times m} \mid \mathbf{P} \mathbf{1}_m = \mathbf{a}, \mathbf{P}^\top \mathbf{1}_n = \mathbf{b} \right\}. \tag{21}$$

*Here, $\mathbf{a}$ and $\mathbf{b}$ are the weight vectors for $\boldsymbol{\alpha}$ and $\boldsymbol{\beta}$.*

The recent research primarily progresses along two paths. The first emphasizes computational efficiency. Exact solutions based on linear programming are often prohibitive for large-scale problems due to their $\mathcal{O}(n^3 \log n)$ complexity where $n$ denotes the number of support points (Bonneel et al., 2011). This limitation has driven the development of approximate methods, including entropic regularization (yielding the Sinkhorn algorithm) with near-quadratic complexity (Altschuler et al., 2017), and sliced optimal transport, which reduces the problem to one-dimensional projections and achieves near-linear complexity. The second path focuses on adapting the OT framework to domain-specific challenges, with applications spanning domain adaptation (Chizat et al., 2018; Courty et al., 2017b), causal inference (Wang et al., 2025a; 2023), missing data imputation (Wang et al., 2025b; 2026a), graph comparison (Xu et al., 2019), recommender system (Liu et al., 2023; 2024a), and generative modeling (Marino and Gerolin, 2020; Chen et al., 2024).

## C  THE COMPUTATION OF CONDITIONAL CORRELATION

Quantifying the autocorrelation structure of the label sequence is challenging due to the confounding effect of the history sequence (Li et al., 2024b;a). Specifically, as discussed in our previous studies (Wang et al., 2025d;c), dependencies among future time steps are difficult to disentangle from spurious correlations stemming from their shared dependence on the historical inputs. It renders standard measures such as the Pearson correlation coefficient unreliable.

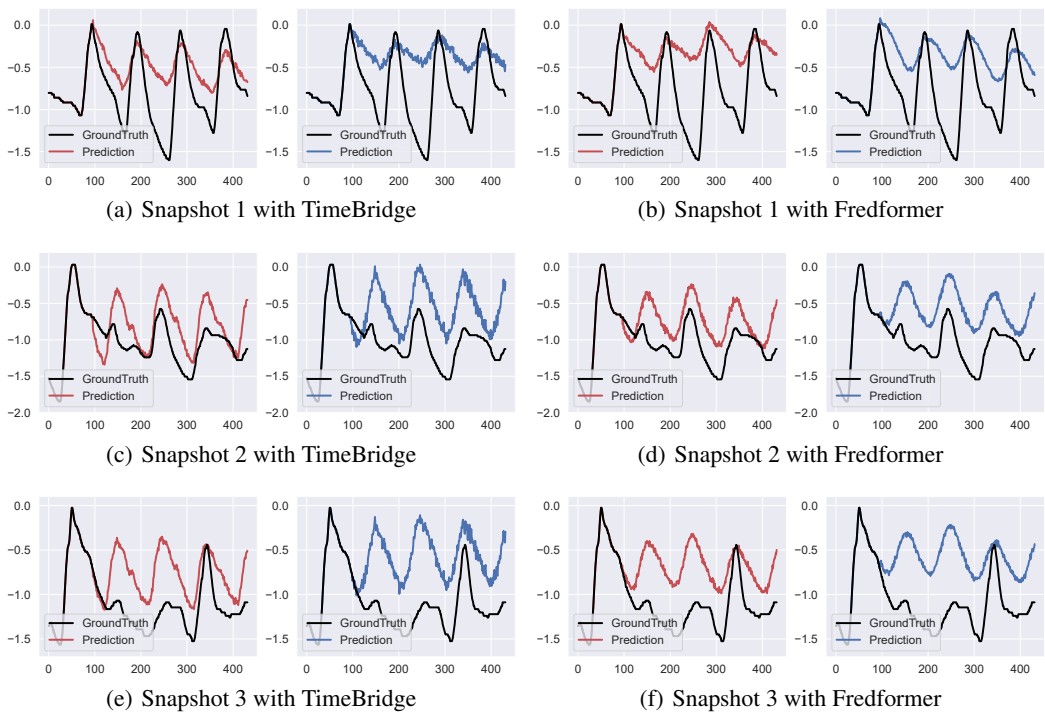

Figure 4: The forecast sequences generated with DF and DistDF. The forecast length is set to 336 and the experiment is conducted on ETTm2.

To address this issue, we employ partial correlation to characterize the autocorrelation structure of the label sequence. It quantifies the correlation between pairs of future time steps while conditioning on the history sequence, thereby removing spurious correlations. The implementation follows standard procedures, as in MATLAB's `partialcorr` function.[6]

## D   MORE EXPERIMENTAL RESULTS

### D.1   COMPARISON WITH PREVIOUS STATE-OF-THE-ART RESULTS

Additional experimental results of comparison against established forecasting models are provided in Table 6. For each dataset, we integrate DistDF with the model achieving the best average performance, to assess both prior state-of-the-art results and the ability of DistDF to further improve them.

### D.2   SHOWCASE

Additional experimental results of showcases are available in Fig. 4 and Fig. 5, where two datasets and two forecasting models are involved.

### D.3   COMPARISON WITH DIFFERENT LEARNING OBJECTIVES

Additional experimental results of learning objective comparison are available in Table 7, where two forecasting models are evaluated across different T values.

---

[6]Implementation is available at https://www.mathworks.com/help/stats/partialcorr.html

Table 6: Comparative results with other established methods for time-series forecasting.

| Models | | DistDF (Ours) | | TimeBridge (2025) | | Fredformer (2024) | | iTransformer (2024) | | FreTS (2023) | | TimesNet (2023) | | MICN (2023) | | TiDE (2023) | | PatchTST (2023) | | DLinear (2023) | |
|---|---|---|---|---|---|---|---|---|---|---|---|---|---|---|---|---|---|---|---|---|---|
| Metrics | | MSE | MAE | MSE | MAE | MSE | MAE | MSE | MAE | MSE | MAE | MSE | MAE | MSE | MAE | MSE | MAE | MSE | MAE | MSE | MAE |
| ETTm1 | 96 | 0.316 | 0.357 | 0.323 | 0.361 | 0.326 | 0.361 | 0.338 | 0.372 | 0.342 | 0.375 | 0.368 | 0.394 | 0.319 | 0.366 | 0.353 | 0.374 | 0.325 | 0.364 | 0.346 | 0.373 |
| | 192 | 0.358 | 0.380 | 0.366 | 0.385 | 0.365 | 0.382 | 0.382 | 0.396 | 0.385 | 0.400 | 0.406 | 0.409 | 0.364 | 0.395 | 0.391 | 0.393 | 0.363 | 0.383 | 0.380 | 0.390 |
| | 336 | 0.392 | 0.404 | 0.398 | 0.408 | 0.396 | 0.404 | 0.427 | 0.424 | 0.416 | 0.421 | 0.454 | 0.444 | 0.395 | 0.425 | 0.423 | 0.414 | 0.404 | 0.413 | 0.413 | 0.414 |
| | 720 | 0.448 | 0.437 | 0.461 | 0.445 | 0.459 | 0.444 | 0.496 | 0.463 | 0.513 | 0.489 | 0.527 | 0.474 | 0.505 | 0.499 | 0.486 | 0.448 | 0.463 | 0.442 | 0.472 | 0.450 |
| | Avg | 0.378 | 0.394 | 0.387 | 0.400 | 0.387 | 0.398 | 0.411 | 0.414 | 0.414 | 0.421 | 0.438 | 0.430 | 0.396 | 0.421 | 0.413 | 0.407 | 0.389 | 0.400 | 0.403 | 0.407 |
| ETTm2 | 96 | 0.174 | 0.256 | 0.177 | 0.259 | 0.177 | 0.260 | 0.182 | 0.265 | 0.188 | 0.279 | 0.184 | 0.262 | 0.178 | 0.277 | 0.182 | 0.265 | 0.180 | 0.266 | 0.188 | 0.283 |
| | 192 | 0.239 | 0.298 | 0.243 | 0.303 | 0.242 | 0.300 | 0.257 | 0.315 | 0.264 | 0.329 | 0.257 | 0.308 | 0.266 | 0.343 | 0.247 | 0.304 | 0.285 | 0.339 | 0.280 | 0.356 |
| | 336 | 0.300 | 0.338 | 0.303 | 0.343 | 0.302 | 0.340 | 0.320 | 0.354 | 0.322 | 0.369 | 0.315 | 0.345 | 0.299 | 0.354 | 0.307 | 0.343 | 0.309 | 0.347 | 0.375 | 0.420 |
| | 720 | 0.397 | 0.394 | 0.401 | 0.399 | 0.399 | 0.397 | 0.423 | 0.411 | 0.489 | 0.482 | 0.452 | 0.421 | 0.489 | 0.482 | 0.408 | 0.398 | 0.437 | 0.422 | 0.526 | 0.508 |
| | Avg | 0.277 | 0.321 | 0.281 | 0.326 | 0.280 | 0.324 | 0.295 | 0.336 | 0.316 | 0.365 | 0.302 | 0.334 | 0.308 | 0.364 | 0.286 | 0.328 | 0.303 | 0.344 | 0.342 | 0.392 |
| ETTh1 | 96 | 0.373 | 0.393 | 0.373 | 0.395 | 0.377 | 0.396 | 0.385 | 0.405 | 0.398 | 0.409 | 0.399 | 0.418 | 0.381 | 0.416 | 0.387 | 0.395 | 0.381 | 0.400 | 0.389 | 0.404 |
| | 192 | 0.428 | 0.425 | 0.428 | 0.426 | 0.437 | 0.425 | 0.440 | 0.437 | 0.451 | 0.442 | 0.452 | 0.451 | 0.497 | 0.489 | 0.439 | 0.425 | 0.450 | 0.443 | 0.442 | 0.440 |
| | 336 | 0.466 | 0.445 | 0.471 | 0.451 | 0.486 | 0.449 | 0.480 | 0.457 | 0.501 | 0.472 | 0.488 | 0.469 | 0.589 | 0.555 | 0.482 | 0.447 | 0.501 | 0.470 | 0.488 | 0.467 |
| | 720 | 0.453 | 0.453 | 0.495 | 0.487 | 0.488 | 0.467 | 0.504 | 0.492 | 0.608 | 0.571 | 0.549 | 0.515 | 0.665 | 0.617 | 0.484 | 0.471 | 0.504 | 0.492 | 0.505 | 0.502 |
| | Avg | 0.430 | 0.429 | 0.442 | 0.440 | 0.447 | 0.434 | 0.452 | 0.448 | 0.489 | 0.474 | 0.472 | 0.463 | 0.533 | 0.519 | 0.448 | 0.435 | 0.459 | 0.451 | 0.456 | 0.453 |
| ETTh2 | 96 | 0.287 | 0.336 | 0.294 | 0.344 | 0.293 | 0.344 | 0.301 | 0.349 | 0.315 | 0.374 | 0.321 | 0.358 | 0.351 | 0.398 | 0.291 | 0.340 | 0.299 | 0.349 | 0.330 | 0.383 |
| | 192 | 0.358 | 0.381 | 0.371 | 0.394 | 0.372 | 0.391 | 0.383 | 0.397 | 0.466 | 0.467 | 0.418 | 0.417 | 0.492 | 0.489 | 0.376 | 0.392 | 0.383 | 0.404 | 0.439 | 0.450 |
| | 336 | 0.408 | 0.421 | 0.421 | 0.429 | 0.420 | 0.433 | 0.425 | 0.432 | 0.522 | 0.502 | 0.464 | 0.454 | 0.656 | 0.582 | 0.417 | 0.427 | 0.439 | 0.444 | 0.589 | 0.538 |
| | 720 | 0.416 | 0.435 | 0.423 | 0.443 | 0.421 | 0.439 | 0.436 | 0.448 | 0.792 | 0.643 | 0.434 | 0.450 | 0.981 | 0.718 | 0.429 | 0.446 | 0.438 | 0.455 | 0.757 | 0.626 |
| | Avg | 0.367 | 0.393 | 0.377 | 0.403 | 0.377 | 0.402 | 0.386 | 0.407 | 0.524 | 0.496 | 0.409 | 0.420 | 0.620 | 0.546 | 0.378 | 0.401 | 0.390 | 0.413 | 0.529 | 0.499 |
| ECL | 96 | 0.137 | 0.235 | 0.142 | 0.239 | 0.161 | 0.258 | 0.150 | 0.242 | 0.180 | 0.266 | 0.170 | 0.272 | 0.170 | 0.281 | 0.197 | 0.274 | 0.170 | 0.264 | 0.197 | 0.282 |
| | 192 | 0.159 | 0.257 | 0.161 | 0.257 | 0.174 | 0.269 | 0.168 | 0.259 | 0.184 | 0.272 | 0.183 | 0.282 | 0.185 | 0.297 | 0.197 | 0.277 | 0.179 | 0.273 | 0.197 | 0.286 |
| | 336 | 0.178 | 0.272 | 0.182 | 0.278 | 0.194 | 0.290 | 0.182 | 0.274 | 0.199 | 0.290 | 0.203 | 0.302 | 0.190 | 0.298 | 0.212 | 0.292 | 0.195 | 0.288 | 0.209 | 0.301 |
| | 720 | 0.212 | 0.302 | 0.217 | 0.309 | 0.235 | 0.319 | 0.214 | 0.304 | 0.234 | 0.322 | 0.294 | 0.366 | 0.221 | 0.329 | 0.254 | 0.325 | 0.234 | 0.320 | 0.245 | 0.334 |
| | Avg | 0.172 | 0.267 | 0.176 | 0.271 | 0.191 | 0.284 | 0.179 | 0.270 | 0.199 | 0.288 | 0.212 | 0.306 | 0.192 | 0.302 | 0.215 | 0.292 | 0.195 | 0.286 | 0.212 | 0.301 |
| Traffic | 96 | 0.380 | 0.262 | 0.391 | 0.268 | 0.461 | 0.327 | 0.397 | 0.271 | 0.531 | 0.323 | 0.590 | 0.316 | 0.498 | 0.298 | 0.646 | 0.386 | 0.444 | 0.284 | 0.649 | 0.397 |
| | 192 | 0.407 | 0.275 | 0.418 | 0.276 | 0.470 | 0.326 | 0.416 | 0.279 | 0.519 | 0.321 | 0.624 | 0.336 | 0.521 | 0.309 | 0.599 | 0.362 | 0.454 | 0.291 | 0.598 | 0.371 |
| | 336 | 0.429 | 0.284 | 0.432 | 0.284 | 0.492 | 0.338 | 0.429 | 0.286 | 0.529 | 0.327 | 0.641 | 0.345 | 0.529 | 0.314 | 0.606 | 0.363 | 0.469 | 0.298 | 0.605 | 0.373 |
| | 720 | 0.452 | 0.297 | 0.464 | 0.301 | 0.521 | 0.353 | 0.462 | 0.303 | 0.573 | 0.346 | 0.670 | 0.356 | 0.567 | 0.326 | 0.643 | 0.383 | 0.506 | 0.319 | 0.646 | 0.395 |
| | Avg | 0.417 | 0.279 | 0.426 | 0.282 | 0.486 | 0.336 | 0.426 | 0.285 | 0.538 | 0.330 | 0.631 | 0.338 | 0.529 | 0.312 | 0.624 | 0.373 | 0.468 | 0.298 | 0.625 | 0.384 |
| Weather | 96 | 0.164 | 0.209 | 0.168 | 0.211 | 0.180 | 0.220 | 0.171 | 0.210 | 0.174 | 0.228 | 0.183 | 0.229 | 0.179 | 0.244 | 0.192 | 0.232 | 0.189 | 0.230 | 0.194 | 0.253 |
| | 192 | 0.212 | 0.252 | 0.214 | 0.254 | 0.222 | 0.258 | 0.246 | 0.278 | 0.213 | 0.266 | 0.242 | 0.276 | 0.242 | 0.310 | 0.240 | 0.270 | 0.228 | 0.262 | 0.238 | 0.296 |
| | 336 | 0.270 | 0.295 | 0.273 | 0.297 | 0.283 | 0.301 | 0.296 | 0.313 | 0.270 | 0.316 | 0.293 | 0.312 | 0.273 | 0.330 | 0.292 | 0.307 | 0.288 | 0.305 | 0.282 | 0.332 |
| | 720 | 0.348 | 0.345 | 0.353 | 0.347 | 0.358 | 0.348 | 0.362 | 0.353 | 0.337 | 0.362 | 0.366 | 0.361 | 0.360 | 0.399 | 0.364 | 0.353 | 0.362 | 0.354 | 0.347 | 0.385 |
| | Avg | 0.248 | 0.275 | 0.252 | 0.277 | 0.261 | 0.282 | 0.269 | 0.289 | 0.249 | 0.293 | 0.271 | 0.295 | 0.264 | 0.321 | 0.272 | 0.291 | 0.267 | 0.288 | 0.265 | 0.317 |
| PEMS03 | 12 | 0.068 | 0.174 | 0.070 | 0.176 | 0.081 | 0.191 | 0.072 | 0.179 | 0.085 | 0.198 | 0.094 | 0.201 | 0.096 | 0.217 | 0.117 | 0.226 | 0.092 | 0.210 | 0.105 | 0.220 |
| | 24 | 0.094 | 0.205 | 0.099 | 0.211 | 0.121 | 0.240 | 0.104 | 0.217 | 0.129 | 0.244 | 0.116 | 0.221 | 0.095 | 0.210 | 0.233 | 0.322 | 0.144 | 0.263 | 0.183 | 0.297 |
| | 36 | 0.116 | 0.229 | 0.126 | 0.240 | 0.180 | 0.292 | 0.137 | 0.251 | 0.173 | 0.286 | 0.134 | 0.237 | 0.107 | 0.223 | 0.379 | 0.418 | 0.200 | 0.309 | 0.258 | 0.361 |
| | 48 | 0.138 | 0.252 | 0.153 | 0.267 | 0.201 | 0.316 | 0.174 | 0.285 | 0.207 | 0.315 | 0.161 | 0.262 | 0.125 | 0.242 | 0.535 | 0.516 | 0.245 | 0.344 | 0.319 | 0.410 |
| | Avg | 0.104 | 0.215 | 0.112 | 0.223 | 0.146 | 0.260 | 0.122 | 0.233 | 0.149 | 0.261 | 0.126 | 0.230 | 0.106 | 0.223 | 0.316 | 0.370 | 0.170 | 0.282 | 0.216 | 0.322 |
| PEMS08 | 12 | 0.076 | 0.177 | 0.080 | 0.184 | 0.091 | 0.199 | 0.084 | 0.187 | 0.096 | 0.205 | 0.111 | 0.208 | 0.161 | 0.274 | 0.121 | 0.233 | 0.106 | 0.223 | 0.113 | 0.225 |
| | 24 | 0.107 | 0.210 | 0.119 | 0.224 | 0.138 | 0.245 | 0.123 | 0.227 | 0.151 | 0.258 | 0.139 | 0.232 | 0.127 | 0.237 | 0.232 | 0.325 | 0.162 | 0.275 | 0.199 | 0.302 |
| | 36 | 0.139 | 0.240 | 0.159 | 0.259 | 0.199 | 0.303 | 0.170 | 0.268 | 0.203 | 0.303 | 0.168 | 0.260 | 0.148 | 0.252 | 0.376 | 0.427 | 0.234 | 0.331 | 0.295 | 0.371 |
| | 48 | 0.171 | 0.265 | 0.198 | 0.289 | 0.255 | 0.338 | 0.218 | 0.306 | 0.247 | 0.334 | 0.189 | 0.272 | 0.175 | 0.270 | 0.543 | 0.527 | 0.301 | 0.382 | 0.389 | 0.429 |
| | Avg | 0.123 | 0.223 | 0.139 | 0.239 | 0.171 | 0.271 | 0.149 | 0.247 | 0.174 | 0.275 | 0.152 | 0.243 | 0.153 | 0.258 | 0.318 | 0.378 | 0.201 | 0.303 | 0.249 | 0.332 |
| 1$^{st}$ Count | | 40 | 41 | 1 | 1 | 0 | 0 | 0 | 0 | 1 | 0 | 0 | 0 | 3 | 2 | 0 | 1 | 0 | 0 | 0 | 0 |

*Note*: In the "DistDF" column, we integrate DistDF with the forecast model that achieves the best average performance on each dataset, to evaluate its ability to further advance state-of-the-art results. We fix the input length as 96 following (Liu et al., 2024b). **Bold** and underlined denote best and second-best results, respectively. *Avg* indicates average results over horizons: T=96, 192, 336 and 720.

## D.4 GENERALIZATION STUDIES

Additional experimental results of varying forecasting models are available in Fig. 6, where four forecasting models are involved on four datasets.

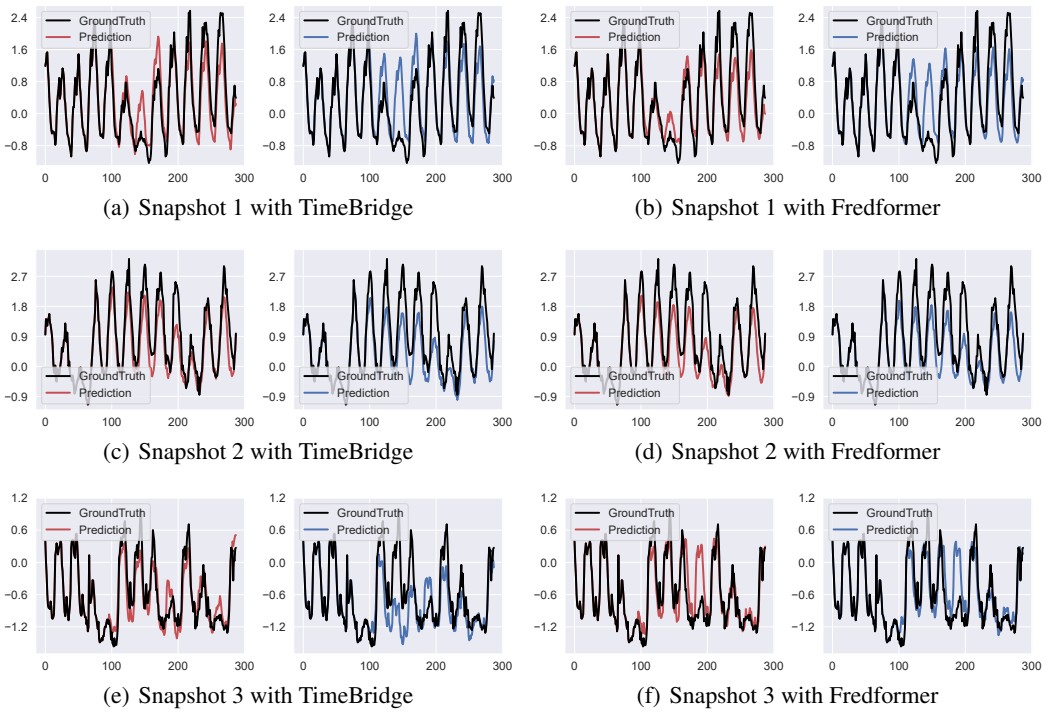

Figure 5: The forecast sequences generated with DF and DistDF. The forecast length is set to 192 and the experiment is conducted on ECL.

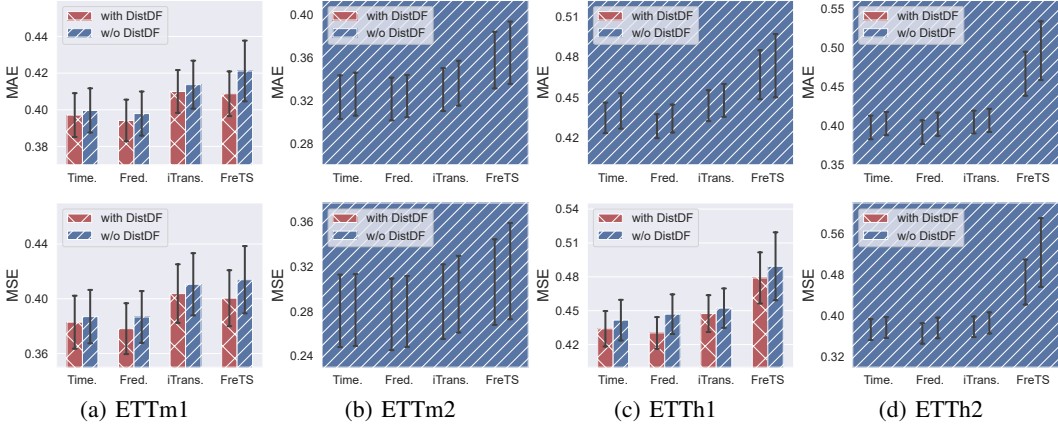

Figure 6: Performance of different forecasting models with and without DistDF. The forecasting errors are averaged over forecast lengths and the error bars represent 50% confidence intervals.

Table 7: Comparable results with different learning objectives.

| Loss | **DistDF** | | Time-o1 | | FreDF | | Koopman | | Dilate | | Soft-DTW | | DF | |
|---|---|---|---|---|---|---|---|---|---|---|---|---|---|---|
| Metrics | MSE | MAE | MSE | MAE | MSE | MAE | MSE | MAE | MSE | MAE | MSE | MAE | MSE | MAE |
| **Forecast model: TimeBridge** | | | | | | | | | | | | | | |
| ETTm1 96 | 0.319 | 0.358 | 0.318 | 0.356 | 0.325 | 0.361 | 0.572 | 0.493 | 0.321 | 0.360 | 0.321 | 0.359 | 0.323 | 0.361 |
| ETTm1 192 | 0.363 | 0.383 | 0.363 | 0.382 | 0.373 | 0.385 | 0.410 | 0.407 | 0.366 | 0.386 | 0.368 | 0.385 | 0.366 | 0.385 |
| ETTm1 336 | 0.394 | 0.405 | 0.396 | 0.407 | 0.398 | 0.406 | 0.397 | 0.408 | 0.397 | 0.409 | 0.405 | 0.410 | 0.398 | 0.408 |
| ETTm1 720 | 0.455 | 0.442 | 0.456 | 0.443 | 0.450 | 0.438 | 0.460 | 0.445 | 0.462 | 0.447 | 0.486 | 0.453 | 0.461 | 0.445 |
| Avg | 0.383 | 0.397 | 0.383 | 0.397 | 0.386 | 0.398 | 0.460 | 0.438 | 0.387 | 0.400 | 0.395 | 0.402 | 0.387 | 0.400 |
| ETTh1 96 | 0.372 | 0.392 | 0.372 | 0.391 | 0.373 | 0.391 | 0.376 | 0.397 | 0.376 | 0.396 | 0.376 | 0.395 | 0.373 | 0.395 |
| ETTh1 192 | 0.424 | 0.429 | 0.422 | 0.423 | 0.425 | 0.421 | 0.426 | 0.430 | 0.430 | 0.433 | 0.425 | 0.427 | 0.428 | 0.426 |
| ETTh1 336 | 0.467 | 0.450 | 0.468 | 0.450 | 0.467 | 0.442 | 0.483 | 0.461 | 0.498 | 0.469 | 0.481 | 0.458 | 0.471 | 0.451 |
| ETTh1 720 | 0.472 | 0.471 | 0.495 | 0.488 | 0.493 | 0.490 | 0.551 | 0.509 | 0.552 | 0.509 | 0.529 | 0.499 | 0.495 | 0.487 |
| Avg | 0.434 | 0.436 | 0.439 | 0.438 | 0.439 | 0.436 | 0.459 | 0.449 | 0.464 | 0.452 | 0.452 | 0.445 | 0.442 | 0.440 |
| ECL 96 | 0.137 | 0.235 | 0.148 | 0.240 | 0.137 | 0.232 | 0.170 | 0.266 | 0.142 | 0.240 | 0.139 | 0.235 | 0.142 | 0.239 |
| ECL 192 | 0.159 | 0.257 | 0.156 | 0.251 | 0.159 | 0.254 | 0.161 | 0.258 | 0.160 | 0.257 | 0.160 | 0.257 | 0.161 | 0.257 |
| ECL 336 | 0.178 | 0.272 | 0.177 | 0.273 | 0.179 | 0.273 | 0.182 | 0.277 | 0.182 | 0.277 | 0.178 | 0.274 | 0.182 | 0.278 |
| ECL 720 | 0.212 | 0.302 | 0.220 | 0.308 | 0.224 | 0.310 | 0.217 | 0.308 | 0.218 | 0.309 | 0.215 | 0.305 | 0.217 | 0.309 |
| Avg | 0.172 | 0.267 | 0.175 | 0.268 | 0.175 | 0.267 | 0.182 | 0.277 | 0.176 | 0.271 | 0.173 | 0.268 | 0.176 | 0.271 |
| Weather 96 | 0.164 | 0.209 | 0.166 | 0.209 | 0.174 | 0.213 | 0.215 | 0.261 | 0.168 | 0.211 | 0.169 | 0.209 | 0.168 | 0.211 |
| Weather 192 | 0.212 | 0.252 | 0.212 | 0.252 | 0.223 | 0.255 | 0.239 | 0.271 | 0.214 | 0.254 | 0.215 | 0.251 | 0.214 | 0.254 |
| Weather 336 | 0.270 | 0.295 | 0.270 | 0.294 | 0.271 | 0.292 | 0.271 | 0.295 | 0.273 | 0.297 | 0.275 | 0.296 | 0.273 | 0.297 |
| Weather 720 | 0.348 | 0.345 | 0.352 | 0.347 | 0.350 | 0.346 | 0.350 | 0.345 | 0.353 | 0.347 | 0.379 | 0.364 | 0.353 | 0.347 |
| Avg | 0.248 | 0.275 | 0.250 | 0.275 | 0.254 | 0.276 | 0.269 | 0.293 | 0.252 | 0.277 | 0.260 | 0.280 | 0.252 | 0.277 |
| **Forecast model: FredFormer** | | | | | | | | | | | | | | |
| ETTm1 96 | 0.316 | 0.357 | 0.321 | 0.357 | 0.326 | 0.355 | 0.335 | 0.368 | 0.337 | 0.367 | 0.332 | 0.363 | 0.326 | 0.361 |
| ETTm1 192 | 0.358 | 0.380 | 0.360 | 0.378 | 0.363 | 0.380 | 0.366 | 0.384 | 0.364 | 0.384 | 0.370 | 0.386 | 0.365 | 0.382 |
| ETTm1 336 | 0.392 | 0.404 | 0.389 | 0.400 | 0.392 | 0.400 | 0.399 | 0.408 | 0.397 | 0.406 | 0.406 | 0.409 | 0.396 | 0.404 |
| ETTm1 720 | 0.448 | 0.437 | 0.447 | 0.435 | 0.455 | 0.440 | 0.456 | 0.441 | 0.457 | 0.443 | 0.478 | 0.450 | 0.459 | 0.444 |
| Avg | 0.378 | 0.394 | 0.379 | 0.393 | 0.384 | 0.394 | 0.389 | 0.400 | 0.389 | 0.400 | 0.397 | 0.402 | 0.387 | 0.398 |
| ETTh1 96 | 0.373 | 0.393 | 0.368 | 0.391 | 0.370 | 0.392 | 0.375 | 0.397 | 0.378 | 0.399 | 0.376 | 0.398 | 0.377 | 0.396 |
| ETTh1 192 | 0.428 | 0.425 | 0.424 | 0.422 | 0.436 | 0.437 | 0.438 | 0.434 | 0.439 | 0.435 | 0.439 | 0.435 | 0.437 | 0.425 |
| ETTh1 336 | 0.466 | 0.445 | 0.467 | 0.441 | 0.473 | 0.443 | 0.473 | 0.455 | 0.481 | 0.453 | 0.484 | 0.455 | 0.486 | 0.449 |
| ETTh1 720 | 0.453 | 0.453 | 0.465 | 0.463 | 0.474 | 0.466 | 0.523 | 0.487 | 0.516 | 0.482 | 0.542 | 0.510 | 0.488 | 0.467 |
| Avg | 0.430 | 0.429 | 0.431 | 0.429 | 0.438 | 0.434 | 0.452 | 0.443 | 0.453 | 0.442 | 0.460 | 0.449 | 0.447 | 0.434 |
| ECL 96 | 0.145 | 0.238 | 0.151 | 0.245 | 0.152 | 0.247 | 0.166 | 0.263 | 0.158 | 0.253 | 0.168 | 0.266 | 0.161 | 0.258 |
| ECL 192 | 0.162 | 0.255 | 0.166 | 0.256 | 0.166 | 0.257 | 0.174 | 0.267 | 0.170 | 0.263 | 0.218 | 0.313 | 0.174 | 0.269 |
| ECL 336 | 0.176 | 0.270 | 0.181 | 0.274 | 0.183 | 0.278 | 0.188 | 0.280 | 0.190 | 0.286 | 0.197 | 0.291 | 0.194 | 0.290 |
| ECL 720 | 0.211 | 0.300 | 0.213 | 0.304 | 0.216 | 0.304 | 0.232 | 0.318 | 0.229 | 0.316 | 0.240 | 0.322 | 0.235 | 0.319 |
| Avg | 0.173 | 0.266 | 0.178 | 0.270 | 0.179 | 0.272 | 0.190 | 0.282 | 0.187 | 0.280 | 0.206 | 0.298 | 0.191 | 0.284 |
| Weather 96 | 0.172 | 0.212 | 0.171 | 0.208 | 0.174 | 0.213 | 0.174 | 0.214 | 0.173 | 0.214 | 0.173 | 0.213 | 0.180 | 0.220 |
| Weather 192 | 0.218 | 0.255 | 0.219 | 0.253 | 0.219 | 0.254 | 0.220 | 0.256 | 0.225 | 0.260 | 0.220 | 0.255 | 0.222 | 0.258 |
| Weather 336 | 0.277 | 0.297 | 0.277 | 0.295 | 0.278 | 0.296 | 0.280 | 0.298 | 0.280 | 0.299 | 0.281 | 0.296 | 0.283 | 0.301 |
| Weather 720 | 0.352 | 0.347 | 0.353 | 0.346 | 0.354 | 0.347 | 0.354 | 0.347 | 0.355 | 0.348 | 0.369 | 0.355 | 0.358 | 0.348 |
| Avg | 0.255 | 0.277 | 0.255 | 0.276 | 0.256 | 0.277 | 0.257 | 0.279 | 0.258 | 0.280 | 0.261 | 0.280 | 0.261 | 0.282 |

## D.5 COMPLEXITY

Additional experimental results of the running time of DistDF are available in Fig. 7. The batch size and dimension are set to 128 and 21, respectively. As the forecast length T increases, the running time for both forward and backward passes generally rises, with some fluctuations. This trend is expected, since T affects the size of the matrices involved in computing the joint-distribution Wasserstein discrepancy in (5). Nevertheless, the running time remains below 1 ms even when T increased to 1024. Furthermore, DistDF's additional computations occur exclusively during training and are completely isolated from the inference stage.

As a result, *DistDF introduces no additional complexity to model inference, and the extra computational cost during training is negligible.*

## D.6 UTILITY TO IMPROVE RECENT FORECASTING MODELS

Additional experimental results demonstrating utility for improving recent forecasting architectures are available in Table 8. We select TQNet (Lin et al., 2025), TimeBridge (Liu et al., 2025), and FredFormer (Piao et al., 2024) as testbeds due to their recency and competitive performance.

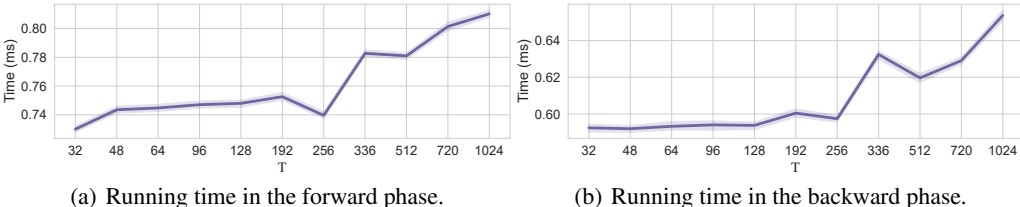

(a) Running time in the forward phase.   (b) Running time in the backward phase.

Figure 7: Running time (ms) with varying forecast length.

### D.7 CONVERGENCE

Additional experimental results on the convergence of the BW discrepancy are available in Fig. 8. The BW objective consistently exhibits a monotonic decrease throughout the training process and reaches a plateau after several epochs, thereby empirically validating the convergence of its optimization. In addition, we examine the evolution of MAE and MSE on the validation set. A significant positive correlation is observed between the dynamics of the BW loss and both forecasting metrics (MAE and MSE). It implies that minimizing the BW discrepancy effectively improves these forecasting metrics.

### D.8 AUTOREGRESSION-BASED FORECASTING PERFORMANCE

Additional experimental results under the autoregression-based forecasting are available in Table 9.

### D.9 PROBABILISTIC FORECASTING PERFORMANCE

Additional experimental results under the probabilistic forecasting setting are available in Table 10, where we select D3U (Li et al., 2025), the state-of-the-art probabilistic forecasting framework as the testbed.

### D.10 MULTI-SCALE FORECASTING PERFORMANCE

Additional experimental results under the multi-scale forecasting setting are available in Table 11, where we select TimeMixer (Wang et al., 2024b) and SCINet (Liu et al., 2022) as the testbeds.

### D.11 CASE STUDY WITH PATCHTST OF VARYING HISTORY LENGTHS

Additional experimental results of varying history lengths are available in Table 12, complementing the fixed length of 96 used in the main text. The forecasting models selected include TimeBridge (Liu et al., 2025) which is the recent state-of-the-art forecasting model, and PatchTST (Nie et al., 2023) which is known to require large history lengths. The results demonstrate that DistDF consistently improves both forecasting models across different history sequence lengths.

### D.12 RANDOM SEED SENSITIVITY

Additional experimental results of random seed sensitivity are available in Table 13, where we report the mean and standard deviation of results obtained from experiments conducted with five different random seeds (2021, 2022, 2023, 2024, and 2025). The results indicate minimal sensitivity of the proposed method to random initialization, as most averaged standard deviations remain below 0.005.

Table 8: The performance comparison of DF and DistDF on different forecasting models.

| Models | | TQNet | | TQNet† | | TimeBridge | | TimeBridge† | | Fredformer | | Fredformer† | | iTransformer | | iTransformer† | | FreTS | | FreTS† |
|---|---|---|---|---|---|---|---|---|---|---|---|---|---|---|---|---|---|---|---|---|
| Metrics | | MSE | MAE | MSE | MAE | MSE | MAE | MSE | MAE | MSE | MAE | MSE | MAE | MSE | MAE | MSE | MAE | MSE | MAE | MSE | MAE |
| ETTh1 | 96 | 0.372 | 0.391 | 0.372 | 0.391 | 0.373 | 0.395 | 0.372 | 0.392 | 0.377 | 0.396 | 0.373 | 0.393 | 0.385 | 0.405 | 0.383 | 0.403 | 0.398 | 0.409 | 0.399 | 0.409 |
| | 192 | 0.430 | 0.424 | 0.430 | 0.422 | 0.428 | 0.426 | 0.424 | 0.429 | 0.437 | 0.425 | 0.428 | 0.425 | 0.440 | 0.437 | 0.438 | 0.434 | 0.451 | 0.442 | 0.457 | 0.447 |
| | 336 | 0.486 | 0.454 | 0.472 | 0.444 | 0.471 | 0.451 | 0.467 | 0.450 | 0.486 | 0.449 | 0.466 | 0.445 | 0.480 | 0.457 | 0.476 | 0.455 | 0.501 | 0.472 | 0.504 | 0.474 |
| | 720 | 0.507 | 0.486 | 0.477 | 0.468 | 0.495 | 0.487 | 0.472 | 0.471 | 0.488 | 0.467 | 0.453 | 0.453 | 0.504 | 0.492 | 0.492 | 0.483 | 0.608 | 0.571 | 0.557 | 0.537 |
| | Avg | 0.449 | 0.439 | 0.438 | 0.431 | 0.442 | 0.440 | 0.434 | 0.436 | 0.447 | 0.434 | 0.430 | 0.429 | 0.452 | 0.448 | 0.447 | 0.444 | 0.489 | 0.474 | 0.479 | 0.467 |
| ETTh2 | 96 | 0.293 | 0.343 | 0.289 | 0.339 | 0.294 | 0.344 | 0.289 | 0.338 | 0.293 | 0.344 | 0.287 | 0.336 | 0.301 | 0.349 | 0.296 | 0.347 | 0.315 | 0.374 | 0.311 | 0.369 |
| | 192 | 0.364 | 0.390 | 0.362 | 0.388 | 0.371 | 0.394 | 0.369 | 0.390 | 0.372 | 0.391 | 0.358 | 0.381 | 0.383 | 0.397 | 0.375 | 0.397 | 0.466 | 0.467 | 0.418 | 0.433 |
| | 336 | 0.411 | 0.424 | 0.410 | 0.424 | 0.421 | 0.429 | 0.415 | 0.426 | 0.420 | 0.433 | 0.408 | 0.421 | 0.425 | 0.432 | 0.421 | 0.434 | 0.522 | 0.502 | 0.521 | 0.505 |
| | 720 | 0.430 | 0.444 | 0.426 | 0.443 | 0.423 | 0.443 | 0.420 | 0.438 | 0.421 | 0.439 | 0.416 | 0.435 | 0.436 | 0.448 | 0.423 | 0.441 | 0.792 | 0.643 | 0.613 | 0.560 |
| | Avg | 0.375 | 0.400 | 0.371 | 0.399 | 0.377 | 0.403 | 0.373 | 0.398 | 0.377 | 0.402 | 0.367 | 0.393 | 0.386 | 0.407 | 0.379 | 0.405 | 0.524 | 0.496 | 0.466 | 0.467 |
| ETTm1 | 96 | 0.310 | 0.352 | 0.311 | 0.351 | 0.323 | 0.361 | 0.319 | 0.358 | 0.326 | 0.361 | 0.316 | 0.357 | 0.338 | 0.372 | 0.334 | 0.372 | 0.342 | 0.375 | 0.335 | 0.371 |
| | 192 | 0.356 | 0.377 | 0.353 | 0.377 | 0.366 | 0.385 | 0.363 | 0.383 | 0.365 | 0.382 | 0.358 | 0.380 | 0.382 | 0.396 | 0.381 | 0.397 | 0.385 | 0.400 | 0.379 | 0.393 |
| | 336 | 0.388 | 0.400 | 0.387 | 0.400 | 0.398 | 0.408 | 0.394 | 0.405 | 0.396 | 0.404 | 0.392 | 0.404 | 0.427 | 0.424 | 0.415 | 0.418 | 0.416 | 0.421 | 0.408 | 0.415 |
| | 720 | 0.450 | 0.437 | 0.449 | 0.436 | 0.461 | 0.445 | 0.455 | 0.442 | 0.459 | 0.444 | 0.448 | 0.437 | 0.496 | 0.463 | 0.485 | 0.454 | 0.513 | 0.489 | 0.479 | 0.456 |
| | Avg | 0.376 | 0.391 | 0.375 | 0.391 | 0.387 | 0.400 | 0.383 | 0.397 | 0.387 | 0.398 | 0.378 | 0.394 | 0.411 | 0.414 | 0.404 | 0.410 | 0.414 | 0.421 | 0.400 | 0.409 |
| ETTm2 | 96 | 0.175 | 0.256 | 0.171 | 0.254 | 0.177 | 0.259 | 0.176 | 0.256 | 0.177 | 0.260 | 0.174 | 0.256 | 0.182 | 0.265 | 0.181 | 0.263 | 0.188 | 0.279 | 0.185 | 0.275 |
| | 192 | 0.243 | 0.300 | 0.234 | 0.295 | 0.243 | 0.303 | 0.241 | 0.300 | 0.242 | 0.300 | 0.239 | 0.298 | 0.257 | 0.315 | 0.249 | 0.307 | 0.264 | 0.329 | 0.253 | 0.318 |
| | 336 | 0.297 | 0.336 | 0.292 | 0.333 | 0.303 | 0.343 | 0.302 | 0.340 | 0.302 | 0.340 | 0.300 | 0.338 | 0.320 | 0.354 | 0.311 | 0.347 | 0.322 | 0.369 | 0.338 | 0.386 |
| | 720 | 0.394 | 0.393 | 0.390 | 0.390 | 0.401 | 0.399 | 0.403 | 0.397 | 0.399 | 0.397 | 0.397 | 0.394 | 0.423 | 0.411 | 0.414 | 0.404 | 0.489 | 0.482 | 0.449 | 0.453 |
| | Avg | 0.277 | 0.321 | 0.272 | 0.318 | 0.281 | 0.326 | 0.280 | 0.323 | 0.280 | 0.324 | 0.277 | 0.321 | 0.295 | 0.336 | 0.289 | 0.330 | 0.316 | 0.365 | 0.306 | 0.358 |
| ECL | 96 | 0.143 | 0.237 | 0.139 | 0.233 | 0.142 | 0.239 | 0.137 | 0.235 | 0.161 | 0.258 | 0.145 | 0.238 | 0.150 | 0.242 | 0.148 | 0.239 | 0.180 | 0.266 | 0.179 | 0.266 |
| | 192 | 0.161 | 0.252 | 0.157 | 0.249 | 0.161 | 0.257 | 0.159 | 0.257 | 0.174 | 0.269 | 0.162 | 0.255 | 0.168 | 0.259 | 0.163 | 0.253 | 0.184 | 0.272 | 0.183 | 0.271 |
| | 336 | 0.178 | 0.270 | 0.174 | 0.267 | 0.182 | 0.278 | 0.178 | 0.272 | 0.194 | 0.290 | 0.176 | 0.270 | 0.182 | 0.274 | 0.176 | 0.270 | 0.199 | 0.290 | 0.199 | 0.288 |
| | 720 | 0.218 | 0.303 | 0.212 | 0.298 | 0.217 | 0.309 | 0.212 | 0.302 | 0.235 | 0.319 | 0.211 | 0.300 | 0.214 | 0.304 | 0.209 | 0.298 | 0.234 | 0.322 | 0.235 | 0.322 |
| | Avg | 0.175 | 0.265 | 0.171 | 0.262 | 0.176 | 0.271 | 0.172 | 0.267 | 0.191 | 0.284 | 0.173 | 0.266 | 0.179 | 0.270 | 0.174 | 0.265 | 0.199 | 0.288 | 0.199 | 0.287 |
| Weather | 96 | 0.160 | 0.203 | 0.160 | 0.202 | 0.168 | 0.211 | 0.164 | 0.209 | 0.180 | 0.220 | 0.172 | 0.212 | 0.171 | 0.210 | 0.174 | 0.214 | 0.174 | 0.228 | 0.173 | 0.229 |
| | 192 | 0.210 | 0.247 | 0.208 | 0.246 | 0.214 | 0.254 | 0.212 | 0.252 | 0.222 | 0.258 | 0.218 | 0.255 | 0.246 | 0.278 | 0.223 | 0.256 | 0.213 | 0.266 | 0.212 | 0.264 |
| | 336 | 0.267 | 0.289 | 0.264 | 0.287 | 0.273 | 0.297 | 0.270 | 0.295 | 0.283 | 0.301 | 0.277 | 0.297 | 0.296 | 0.313 | 0.280 | 0.299 | 0.270 | 0.316 | 0.263 | 0.305 |
| | 720 | 0.346 | 0.342 | 0.344 | 0.342 | 0.353 | 0.347 | 0.348 | 0.345 | 0.358 | 0.348 | 0.352 | 0.347 | 0.362 | 0.353 | 0.357 | 0.350 | 0.337 | 0.362 | 0.331 | 0.355 |
| | Avg | 0.246 | 0.270 | 0.244 | 0.269 | 0.252 | 0.277 | 0.248 | 0.275 | 0.261 | 0.282 | 0.255 | 0.277 | 0.269 | 0.289 | 0.258 | 0.280 | 0.249 | 0.293 | 0.245 | 0.288 |

*Note*: The length of history window is set to 96 for all baselines. `Avg` indicates the results averaged over forecasting lengths: T=96, 192, 336 and 720. † marks the forecasting model trained via DistDF.

Table 9: The performance comparison of DF and DistDF on the autoregressive forecasting setting.

| Models | | TimeBridge | | TimeBridge† | | Fredformer | | Fredformer† | |
|---|---|---|---|---|---|---|---|---|---|
| Metrics | | MSE | MAE | MSE | MAE | MSE | MAE | MSE | MAE |
| ETTm1 | 96 | 0.405 | 0.402 | 0.395 | 0.391 | 0.391 | 0.396 | 0.386 | 0.390 |
| | 192 | 0.467 | 0.438 | 0.419 | 0.408 | 0.494 | 0.449 | 0.493 | 0.446 |
| | 336 | 0.518 | 0.467 | 0.460 | 0.437 | 0.572 | 0.500 | 0.579 | 0.486 |
| | 720 | 0.725 | 0.514 | 0.527 | 0.478 | 1.821 | 0.837 | 0.833 | 0.563 |
| | Avg | 0.528 | 0.455 | 0.450 | 0.428 | 0.820 | 0.546 | 0.573 | 0.471 |
| Weather | 96 | 0.527 | 0.343 | 0.241 | 0.275 | 0.241 | 0.267 | 0.211 | 0.245 |
| | 192 | 1.165 | 0.494 | 0.303 | 0.320 | 0.306 | 0.318 | 0.274 | 0.292 |
| | 336 | 4.826 | 0.749 | 0.371 | 0.365 | 0.330 | 0.331 | 0.312 | 0.322 |
| | 720 | 9.363 | 1.374 | 0.461 | 0.421 | 0.433 | 0.406 | 0.407 | 0.380 |
| | Avg | 3.970 | 0.740 | 0.344 | 0.345 | 0.327 | 0.330 | 0.301 | 0.310 |

*Note*: The length of history window is set to 96 for all baselines. `Avg` indicates the results averaged over forecasting lengths: T=96, 192, 336 and 720. † marks the forecasting model trained via DistDF.

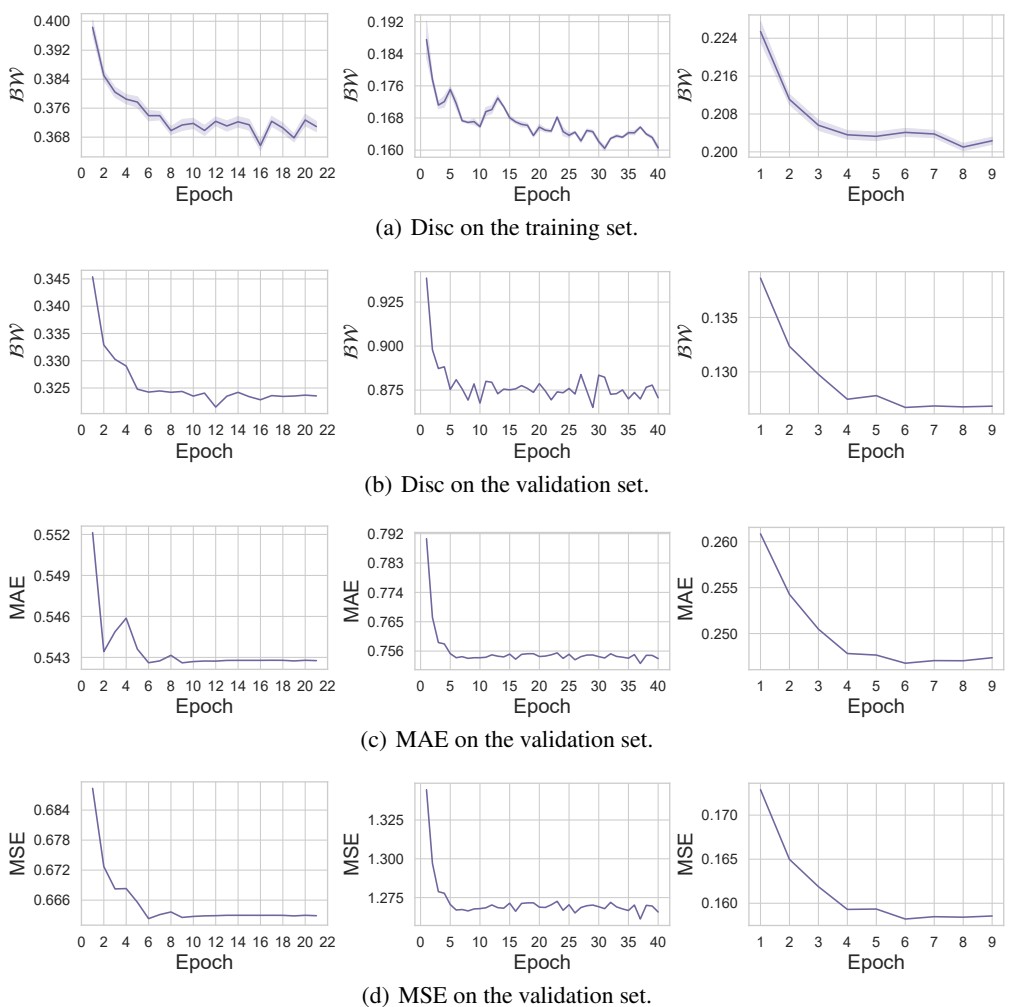

(a) Disc on the training set.

(b) Disc on the validation set.

(c) MAE on the validation set.

(d) MSE on the validation set.

Figure 8: Evolution of training objectives and validation metrics across four datasets: ETTm1, ETTh1, and ECL (from left to right).

Table 10: The performance comparison of DF and DistDF on the probabilistic forecasting task.

| Models | | D3U | | | | D3U$^\dagger$ | | | |
|---|---|---|---|---|---|---|---|---|---|
| Metrics | | MSE | MAE | CRPS | CRPS$_{sum}$ | MSE | MAE | CRPS | CRPS$_{sum}$ |
| ETTm1 | 96 | 0.317 | 0.357 | 0.263 | 0.723 | 0.316 | 0.357 | 0.265 | 0.720 |
| | 192 | 0.361 | 0.383 | 0.285 | 0.749 | 0.360 | 0.383 | 0.282 | 0.747 |
| | 336 | 0.394 | 0.404 | 0.299 | 0.742 | 0.390 | 0.402 | 0.298 | 0.731 |
| | 720 | 0.460 | 0.437 | 0.325 | 0.892 | 0.453 | 0.435 | 0.328 | 0.849 |
| | Avg | 0.383 | 0.395 | 0.293 | 0.776 | 0.380 | 0.394 | 0.293 | 0.762 |
| Weather | 96 | 0.176 | 0.240 | 0.174 | 0.179 | 0.173 | 0.225 | 0.171 | 0.173 |
| | 192 | 0.223 | 0.271 | 0.205 | 0.234 | 0.217 | 0.265 | 0.198 | 0.210 |
| | 336 | 0.279 | 0.309 | 0.233 | 0.269 | 0.278 | 0.310 | 0.233 | 0.260 |
| | 720 | 0.359 | 0.361 | 0.273 | 0.419 | 0.353 | 0.360 | 0.269 | 0.378 |
| | Avg | 0.259 | 0.295 | 0.221 | 0.275 | 0.255 | 0.290 | 0.218 | 0.255 |

*Note*: The length of history window is set to 96 for all baselines. Avg indicates the results averaged over forecasting lengths: T=96, 192, 336 and 720.
$^\dagger$ marks the forecasting model trained via DistDF.

Table 11: The performance comparison of DF and DistDF on the multi-scale architectures.

| Models | | TimeMixer | | TimeMixer$^\dagger$ | | SCINet | | SCINet$^\dagger$ | |
|---|---|---|---|---|---|---|---|---|---|
| Metrics | | MSE | MAE | MSE | MAE | MSE | MAE | MSE | MAE |
| ETTm1 | 96 | 0.329 | 0.369 | 0.326 | 0.369 | 0.325 | 0.365 | 0.319 | 0.359 |
| | 192 | 0.371 | 0.391 | 0.373 | 0.392 | 0.383 | 0.397 | 0.367 | 0.385 |
| | 336 | 0.427 | 0.425 | 0.412 | 0.423 | 0.436 | 0.424 | 0.403 | 0.406 |
| | 720 | 0.564 | 0.506 | 0.491 | 0.459 | 0.528 | 0.476 | 0.469 | 0.444 |
| | Avg | 0.422 | 0.423 | 0.401 | 0.411 | 0.418 | 0.416 | 0.389 | 0.399 |
| ETTh1 | 96 | 0.419 | 0.426 | 0.400 | 0.410 | 0.409 | 0.415 | 0.397 | 0.405 |
| | 192 | 0.464 | 0.451 | 0.439 | 0.436 | 0.457 | 0.441 | 0.448 | 0.434 |
| | 336 | 0.509 | 0.472 | 0.485 | 0.450 | 0.499 | 0.461 | 0.491 | 0.455 |
| | 720 | 0.614 | 0.553 | 0.501 | 0.486 | 0.505 | 0.482 | 0.501 | 0.479 |
| | Avg | 0.501 | 0.476 | 0.456 | 0.446 | 0.467 | 0.450 | 0.459 | 0.443 |
| ECL | 96 | 0.159 | 0.260 | 0.145 | 0.242 | 0.146 | 0.248 | 0.141 | 0.242 |
| | 192 | 0.161 | 0.258 | 0.159 | 0.256 | 0.167 | 0.266 | 0.159 | 0.257 |
| | 336 | 0.173 | 0.272 | 0.176 | 0.272 | 0.179 | 0.280 | 0.177 | 0.277 |
| | 720 | 0.212 | 0.302 | 0.207 | 0.298 | 0.202 | 0.298 | 0.197 | 0.294 |
| | Avg | 0.176 | 0.273 | 0.172 | 0.267 | 0.173 | 0.273 | 0.169 | 0.268 |
| Weather | 96 | 0.173 | 0.220 | 0.168 | 0.217 | 0.160 | 0.208 | 0.158 | 0.207 |
| | 192 | 0.213 | 0.254 | 0.212 | 0.253 | 0.214 | 0.257 | 0.211 | 0.254 |
| | 336 | 0.286 | 0.306 | 0.273 | 0.298 | 0.276 | 0.300 | 0.271 | 0.298 |
| | 720 | 0.377 | 0.362 | 0.354 | 0.352 | 0.362 | 0.356 | 0.359 | 0.351 |
| | Avg | 0.262 | 0.285 | 0.252 | 0.280 | 0.253 | 0.280 | 0.250 | 0.278 |

*Note*: The length of history window is set to 96 for all baselines. `Avg` indicates the results averaged over forecasting lengths: T=96, 192, 336 and 720. $^\dagger$ marks the forecasting model trained via DistDF.

Table 12: Varying input sequence length results on the Weather dataset.

| Models | | | **DistDF** | | TimeBridge | | **DistDF** | | PatchTST | |
|---|---|---|---|---|---|---|---|---|---|---|
| Metrics | | | MSE | MAE | MSE | MAE | MSE | MAE | MSE | MAE |
| Historical sequence length | 96 | 96 | 0.164 | 0.209 | 0.168 | 0.211 | 0.179 | 0.220 | 0.189 | 0.230 |
| | | 192 | 0.212 | 0.252 | 0.214 | 0.254 | 0.222 | 0.257 | 0.228 | 0.262 |
| | | 336 | 0.270 | 0.295 | 0.273 | 0.297 | 0.278 | 0.298 | 0.288 | 0.305 |
| | | 720 | 0.348 | 0.345 | 0.353 | 0.347 | 0.354 | 0.348 | 0.362 | 0.354 |
| | | Avg | 0.248 | 0.275 | 0.252 | 0.277 | 0.258 | 0.281 | 0.267 | 0.288 |
| | 192 | 96 | 0.160 | 0.207 | 0.163 | 0.210 | 0.157 | 0.203 | 0.163 | 0.209 |
| | | 192 | 0.202 | 0.244 | 0.205 | 0.248 | 0.202 | 0.244 | 0.207 | 0.249 |
| | | 336 | 0.260 | 0.290 | 0.259 | 0.288 | 0.258 | 0.285 | 0.268 | 0.293 |
| | | 720 | 0.335 | 0.342 | 0.338 | 0.344 | 0.335 | 0.338 | 0.511 | 0.451 |
| | | Avg | 0.239 | 0.271 | 0.241 | 0.273 | 0.238 | 0.267 | 0.287 | 0.301 |
| | 336 | 96 | 0.155 | 0.206 | 0.156 | 0.206 | 0.153 | 0.204 | 0.158 | 0.208 |
| | | 192 | 0.198 | 0.244 | 0.199 | 0.245 | 0.200 | 0.249 | 0.235 | 0.291 |
| | | 336 | 0.245 | 0.283 | 0.259 | 0.294 | 0.250 | 0.285 | 0.252 | 0.287 |
| | | 720 | 0.325 | 0.337 | 0.323 | 0.335 | 0.323 | 0.337 | 0.326 | 0.336 |
| | | Avg | 0.231 | 0.267 | 0.234 | 0.270 | 0.232 | 0.269 | 0.243 | 0.280 |
| | 720 | 96 | 0.147 | 0.198 | 0.148 | 0.201 | 0.149 | 0.204 | 0.153 | 0.205 |
| | | 192 | 0.197 | 0.247 | 0.203 | 0.253 | 0.196 | 0.247 | 0.205 | 0.254 |
| | | 336 | 0.240 | 0.279 | 0.239 | 0.278 | 0.247 | 0.291 | 0.248 | 0.288 |
| | | 720 | 0.319 | 0.339 | 0.329 | 0.346 | 0.313 | 0.333 | 0.317 | 0.339 |
| | | Avg | 0.226 | 0.266 | 0.230 | 0.269 | 0.226 | 0.269 | 0.231 | 0.272 |

Table 13: Experimental results ($\text{mean}_{\pm\text{std}}$) with varying seeds (2021-2025).

| Dataset | | ECL | | | | Weather | | | |
|---|---|---|---|---|---|---|---|---|---|
| Models | | **DistDF** | | DF | | **DistDF** | | DF | |
| Metrics | | MSE | MAE | MSE | MAE | MSE | MAE | MSE | MAE |
| 96 | | $0.138_{\pm0.001}$ | $0.236_{\pm0.001}$ | $0.141_{\pm0.001}$ | $0.239_{\pm0.001}$ | $0.167_{\pm0.003}$ | $0.209_{\pm0.001}$ | $0.169_{\pm0.001}$ | $0.212_{\pm0.001}$ |
| 192 | | $0.159_{\pm0.001}$ | $0.257_{\pm0.001}$ | $0.161_{\pm0.000}$ | $0.258_{\pm0.001}$ | $0.213_{\pm0.001}$ | $0.253_{\pm0.001}$ | $0.215_{\pm0.001}$ | $0.254_{\pm0.001}$ |
| 336 | | $0.179_{\pm0.001}$ | $0.272_{\pm0.001}$ | $0.183_{\pm0.002}$ | $0.279_{\pm0.002}$ | $0.271_{\pm0.002}$ | $0.296_{\pm0.002}$ | $0.272_{\pm0.001}$ | $0.296_{\pm0.001}$ |
| 720 | | $0.210_{\pm0.001}$ | $0.301_{\pm0.001}$ | $0.221_{\pm0.005}$ | $0.311_{\pm0.004}$ | $0.349_{\pm0.002}$ | $0.347_{\pm0.002}$ | $0.352_{\pm0.002}$ | $0.348_{\pm0.001}$ |
| Avg | | $0.172_{\pm0.000}$ | $0.266_{\pm0.001}$ | $0.177_{\pm0.002}$ | $0.272_{\pm0.001}$ | $0.250_{\pm0.001}$ | $0.276_{\pm0.001}$ | $0.252_{\pm0.001}$ | $0.277_{\pm0.000}$ |

