# OpenReview forum: "DistDF: Time-series Forecasting Needs Joint-distribution Wasserstein Alignment"
_ICLR.cc/2026/Conference — ICLR 2026 Poster_

### Official Review · Reviewer_6qSb · 2025-10-31

**Soundness:** 3
**Presentation:** 4
**Contribution:** 3
**Rating:** 6
**Confidence:** 3

**Summary:**

This paper proposes Distribution-aware Direct Forecast (DistDF), which achieves alignment by minimizing joint-distribution Wasserstein discrepancy between conditional forecast and label distributions to enhance forecast accuracy.

**Strengths:**

1. This paper is well written and polished. Notations and equations are clearly presented and explained.
2. This paper is well-motivated and offers an extremely thorough explanation.
3. Experiments are comprehensive.

**Weaknesses:**

1. Experimental comparison (Tab. 2) lacks some most recent works, e.g., [*1]. The proposed method might not outperform these new works. TQNet [*1] achieves **0.377** MSE, **0.393** MAE on ETTm1.
2. Experimental results could not fully support the significance of the method. The improvement is marginal when compared to prior art, e.g., TimeBridge, Time-o1, and TQNet [*1].
3. Improvement on presentation:
 - For results in the table, should not use **Bold** and $\underline{\text{Underline}}$ when two numbers are the same, use Bold for both.
 - (minor) In Section 4.3, the reference to Table 4 should be changed to Table 2.
 - (minor) Use consistent table style. Use \toprule for Tab. 5 & 6


[*1] Lin, Shengsheng, et al. "Temporal Query Network for Efficient Multivariate Time Series Forecasting." Forty-second International Conference on Machine Learning.

**Questions:**

See weaknesses.

---

> ### Author Response · Authors · 2025-11-27
>
> Thank you so much for your encouraging evaluation and appreciation of our **presentation, motivation, and empirical studies**. Below are our responses to the specific query raised.
>
> #### [W1] Experimental comparison (Tab. 2) lacks some most recent works, e.g., [1]. The proposed method might not outperform these new works. **TQNet** [*1] achieves 0.377 MSE, 0.393 MAE on ETTm1.
>
> **Response.** Thank you very much for your meticulous observation. Our response to this query is structured as follows:
>
> - Firstly, we would clarify that as a learning objective, the performance of DistDF is highly impacted by the forecasting model used. Therefore, to investigate the applicability of DistDF over TQNet, we should compare TQNet trained with MSE and TQNet trained with DistDF.
> - **Additional experiments.** We add experiments to evaluate how DistDF performs when specifying TQNet and TimeBridge as the forecasting models. The results are available in the table below, where TQNet$^\dagger$ and TimeBridge$^\dagger$ are trained via DistDF. Overall, TQNet$^\dagger$ and TimeBridge$^\dagger$ consistently outperform TQNet and TimeBridge, respectively, **validating the utility of DistDF over state-of-the-art forecasting models.**
> - **Revision.** We add the table below in the revised manuscript (Table 15), to demonstrate the efficacy of DistDF to improve TQNet and TimeBridge.
>
>
> | Dataset | TQNet (MSE) | TQNet (MAE) | TQNet$^\dagger$ (MSE) | TQNet$^\dagger$ (MAE) | TimeBridge (MSE) | TimeBridge (MAE) | TimeBridge$^\dagger$ (MSE) | TimeBridge$^\dagger$ (MAE) |
> | :--- | :---: | :---: | :---: | :---: | :---: | :---: | :---: | :---: |
> | **ETTh1** | 0.449 | 0.439 | 0.438 | 0.431 | 0.442 | 0.440 | 0.434 | 0.436 |
> | **ETTh2** | 0.375 | 0.400 | 0.371 | 0.399 | 0.377 | 0.403 | 0.373 | 0.398 |
> | **ETTm1** | 0.376 | 0.391 | 0.375 | 0.391 | 0.387 | 0.400 | 0.383 | 0.397 |
> | **ETTm2** | 0.277 | 0.321 | 0.272 | 0.318 | 0.281 | 0.326 | 0.280 | 0.323 |
> | **ECL** | 0.175 | 0.265 | 0.171 | 0.262 | 0.176 | 0.271 | 0.172 | 0.267 |
> | **Weather** | 0.246 | 0.270 | 0.244 | 0.269 | 0.252 | 0.277 | 0.248 | 0.275 |
>
> #### [W2] Experimental results could not fully support the significance of the method. The improvement is marginal when compared to prior art, e.g., **TimeBridge, Time-o1, and TQNet [*1].**
>
> **Response.** Thank you very much for your detailed and meticulous comment. We would like to address this concern as follows.
>
> - Firstly, we would clarify that **DistDF can effectively improve the performance of different architectures**, such as **TimeBridge and TQNet**, as discussed in the response to [W1].
> - Secondly, we recognize that compared to Time-o1, the performance improvement achieved by DistDF is **moderate (yet consistent)**. However, it is important to emphasize that **with respect to the widespread direct forecasting framework based on MSE optimization, DistDF yields substantial gains**, which provides empirical validation on the benefit of the proposed discrepancy-based objective to enhance forecasting performance. Furthermore, compared to Time-o1, DistDF contributes beyond empirical improvements in two key aspects: (i) it **identifies an inherent theoretical limitation** of the Time-o1 learning objective; and (ii) it **reformulates time-series forecasting as a distribution alignment challenge**, underscoring the importance of distribution alignment. This establishes conceptual links to research areas such as transfer learning, where distribution alignment is extensively developed, which brings new insights to the time-series forecasting field.
>
> #### **[W3] Improvement on presentation. (i) For results in the table, should not use Bold and Underline when two numbers are the same, use Bold for both. (ii) In Section 4.3, the reference to Table 4 should be changed to Table 2. (iii) Use consistent table style. Use \toprule for Tab. 5 & 6**
>
> **Response.** Thank you very much for your meticulous comment! We have revised these typographical errors and improved the format in the revised manuscript.
> - For results in the table, when two reported values are identical up to the first three decimal places but differ in subsequent digits (e.g., 0.3241 vs. 0.3244), we boldface the numerically superior value based on the full reported precision. This clarification has been added to the associated tablenotes in the revised manuscript: `When metric values coincide up to three decimal places, Bold indicates the numerically superior result based on full precision.`
> - We have changed the reference to Table 4 to Table 2 in the revised manuscript.
> - We have used \toprule for Tab. 5 & 6 in the revised manuscript.

---

### Official Review · Reviewer_c2pc · 2025-10-31

**Soundness:** 2
**Presentation:** 2
**Contribution:** 2
**Rating:** 4
**Confidence:** 4

**Summary:**

This paper proposes DistDF, a new training objective for time-series forecasting that aims to align the conditional distributions of forecasts and labels, rather than relying on point-wise MSE. Since conditional discrepancies are difficult to estimate from limited data, the authors introduce a joint-distribution Wasserstein discrepancy, optimized between the distributions of (history, labels) and (history, predictions). The method is model-agnostic and can be plugged into existing forecasting models. Experiments show performance improvements on multiple benchmarks.

**Strengths:**

- Strong and clearly articulated motivation regarding autocorrelation bias in likelihood-based objectives.
- Solid theoretical foundation, including alignment guarantees and non-negativity properties of the objective.
- Method is architecture-agnostic, enabling integration with a broad range of forecasting models.
- Extensive benchmarking shows consistent improvements, supported by ablation studies demonstrating contribution of components.
- Generally clear writing and clean presentation of the mathematical formulation.

**Weaknesses:**

- A key limitation is that the proposed discrepancy objective lacks guaranteed convergence or clear interpretability during training, making its practical effect on conditional alignment somewhat uncertain. Because the loss must be combined with MSE, the discrepancy may act more like a regularizer than a principled stand-alone objective. Additional empirical analysis of its optimization dynamics and correlation with performance would strengthen the claims.
- More comprehensive experiments are needed to isolate the contribution of the proposed objective. Given that the method relies on a weighted combination with MSE, it should be compared not only against plain MSE training but also against other established time-series learning objectives (e.g., Dilate, Soft-DTW) when similarly combined with MSE. Such comparisons would help determine whether the observed gains stem from the specific discrepancy formulation or simply from augmenting the loss with an auxiliary term.
- Evaluation is restricted to direct forecasting, limiting evidence of robustness across different training paradigms. Additional experiments under an autoregressive setting would be valuable to validate whether the proposed objective is broadly applicable across different forecasting architectures and training pipelines.
- In Table 1, it is unclear which underlying model architectures DistDF is applied to. Since DistDF is a learning objective rather than a new architecture, and the table compares against architectural baselines, the presentation may confuse readers regarding what is being evaluated. Clarifying the base model used for each dataset would improve readability. Explicitly specifying the base architecture for each dataset (e.g., as done in Scaleformer, ICLR 2023) would improve clarity and ensure a fair interpretation of the reported gains.
ref. Scaleformer: Iterative Multi-scale Refining Transformers for Time Series Forecasting, ICLR 2023

**Questions:**

- Since the proposed objective must be combined with MSE for stable training, can the authors provide evidence that the improvement does not simply arise from a regularization effect? For example, how does the discrepancy term alone behave, and how strongly does its reduction correlate with forecasting accuracy?
- The distinction between DistDF and existing learning-objective methods such as Time-o1, FreDF, Koopman-based losses, and Soft-DTW remains somewhat unclear. Can the authors more explicitly highlight the conceptual and practical differences, particularly regarding theoretical guarantees and optimization behavior?
- The discussion of likelihood bias focuses primarily on MSE. Do similar issues arise in probabilistic forecasting frameworks using alternative objectives (e.g., quantile loss, CRPS)? If so, is DistDF compatible with or beneficial under such setups?
- How well does DistDF extend to multivariate forecasting, probabilistic output formulations, or multi-scale architectures? Providing results or analysis in these more general settings would help verify that the proposed approach is broadly applicable beyond the current scope.

---

> ### Author Response · Authors · 2025-11-27
>
> We sincerely appreciate the reviewer for the meticulous comments and appreciation of our **motivation**, **theoretical foundation**, **implementation**, **benchmarking** and **presentation**. We have taken every effort to address the raised concerns through additional explanations and experiments. Below are our responses to the specific query raised.
>
> ---
>
> #### [W1] A key limitation is that the proposed discrepancy objective **lacks guaranteed convergence** or clear interpretability during training, making its practical effect on conditional alignment somewhat uncertain. Because the loss must be combined with MSE, the discrepancy may act more like a regularizer than a principled stand-alone objective. Additional empirical analysis of **optimization dynamics** and **correlation with performance** would strengthen the claims.
>
> **Response.** Thank you for your thoughtful comment. We agree that it is essential to discuss the convergence, optimization dynamics, and their correlation with performance. We structure our response as follows.
> - Firstly, **we add experiments to discuss the convergence** of BW discrepancy and **analyze its optimization dynamics**.
>   - **Setup.** Using BW discrepancy as the learning objective, we monitored its trajectory across epochs on the ECL, ETTh1, and ETTm1 datasets. The results are illustrated below.
>   - **Result analysis.** **The optimization dynamics confirm the convergence of the BW discrepancy**: (i) the loss value decreases consistently throughout the training process; and (ii) it stabilizes at a specific equilibrium with minimal fluctuation after several epochs.
> | Epoch  | 1 | 2 | 3 | 4 | 5 | 6 | 7 | 8 | 9 | 10 |
> |-|--|--|--|--|--|--|--|--|--|-|
> | ECL  | 0.287  | 0.229  | 0.225  | 0.217  | 0.219  | 0.207  | 0.201  | 0.208  | 0.201  | 0.201 |
> | ETTh1  | 0.209  | 0.172  | 0.175  | 0.168  | 0.166  | 0.169  | 0.170  | 0.166  | 0.164  | 0.164 |
> | ETTm1  | 0.461  | 0.421  | 0.385  | 0.407  | 0.372  | 0.394  | 0.367  | 0.379  | 0.372  | 0.382 |
> - Secondly, **we add experiments to discuss the correlation between performance and BW dynamics.**
>   - **Setup.** Following the above setup, we further track the trajectory of MAE and MSE on the validation set. Then, we calculate the Pearson correlation coefficient between the BW discrepancy trajectory on the training and validation sets and the MAE/MSE trajectories on the validation set. This quantifies the correlation between the BW discrepancy and the performance metrics. The results are illustrated below.
>   - **Result analysis.** **The results confirm the positive correlation between the BW discrepancy and the performance.** In particular, the correlation coefficient between the BW discrepancy trajectory on the training set and the MSE trajectory on the validation set is 0.999, which indicates a strong positive correlation. This implies that minimizing the BW discrepancy can effectively improve the MAE and MSE performance.
> | Correlation coef. | MSE (valid.) | MAE (valid.) |
> |-|-|--|
> | ECL |
> | BW (train.) | 0.968| 0.952|
> | BW (valid.) | 0.999| 0.995|
> | ETTh1 |
> | BW (training) | 0.534| 0.536|
> | BW (validation) | 0.856| 0.858|
> | ETTm1 |
> | BW (training) | 0.891| 0.747|
> | BW (validation) | 0.972| 0.862|
>
> - **Revision.** We add the trajectories of the BW discrepancy, MAE, and MSE in Figure 8, accompanied by a detailed discussion in Appendix D.10.
>
> #### [W2] More comprehensive experiments are needed to isolate the contribution of the proposed objective. Given that the method relies on a weighted combination with MSE, it should be compared not only against plain MSE training **but also compared against other established time-series learning objectives (e.g., Dilate, Soft-DTW) when similarly combined with MSE**. Such comparisons would help determine whether the observed gains stem from the specific discrepancy formulation or simply from augmenting the loss with an auxiliary term.
>
> **Response.** We agree that when comparing with other objectives, it is necessary to combine them with MSE, which is exactly what we did. Please see our concise responses below.
> - Firstly, we clarify that **in our comparisons with other objectives (e.g., FreDF, Dilate, Soft-DTW) in Table 2, we have consistently combined these objectives with MSE,** incorporating other objectives as regularizers. This protocol follows the precedent set by [1] and ensures a fair and consistent evaluation across methods.
> - Secondly, **we add experimental results where these objectives are employed without  the MSE loss.** The reported results are averaged over four forecast horizons (96, 192, 336, 720). Overall, **the performances degrade compared to those reported in the manuscript.** For instance, on the ETTh1 dataset with the TimeBridge forecasting model, the best-performing Time-o1 objective achieves an MSE of 0.451 and an MAE of 0.441 when used alone, whereas when combined with MSE as in our main manuscript, the corresponding values improve to MSE 0.434 and MAE 0.436.

---

> ### Author Response · Authors · 2025-11-27
>
> |           |        | Time-o1|      | FreDF|        | Koopman |     | Dilate|       | LDTW  |       | Soft-DTW |    | DTW  |        | DF  |         |
> |-----|--------|--|--|--|--|--|--|--|--|--|--|--|--|--|--|--|--|
> |       |    | MSE  | MAE   | MSE  | MAE   | MSE  | MAE   | MSE  | MAE   | MSE  | MAE   | MSE  | MAE   | MSE  | MAE   | MSE  | MAE   |
> | **TimeBridge** | ETTm1  | 0.386 | 0.394 | 0.398 | 0.396 | 0.543 | 0.489 | 0.937 | 0.632 | 0.387 | 0.400 | 1.047 | 0.637 | 0.620 | 0.488 | 0.387 | 0.400 |
> |           | ETTh1  | 0.451 | 0.441 | 0.443 | 0.438 | 0.757 | 0.593 | 0.915 | 0.650 | 0.464 | 0.452 | 0.971 | 0.632 | 0.879 | 0.597 | 0.442 | 0.440 |
> |           | ECL    | 0.172 | 0.262 | 0.336 | 0.371 | 7.845 | 1.577 | 1.124 | 0.841 | 0.352 | 0.402 | 1.252 | 0.824 | 1.162 | 0.787 | 0.176 | 0.271 |
> |           | Weather| 0.255 | 0.276 | 0.256 | 0.277 | 0.283 | 0.303 | 0.329 | 0.340 | 0.252 | 0.277 | 0.305 | 0.311 | 0.264 | 0.283 | 0.252 | 0.277 |
> | **Fredformer** | ETTm1  | 0.385 | 0.393 | 0.385 | 0.393 | 0.467 | 0.449 | 0.719 | 0.564 | 0.389 | 0.400 | 0.735 | 0.558 | 0.637 | 0.498 | 0.387 | 0.398 |
> |           | ETTh1  | 0.431 | 0.429 | 0.438 | 0.434 | 0.571 | 0.513 | 0.715 | 0.577 | 0.477 | 0.457 | 0.870 | 0.609 | 0.832 | 0.585 | 0.447 | 0.434 |
> |           | ECL    | 0.178 | 0.270 | 0.179 | 0.271 | 0.399 | 0.462 | 0.864 | 0.766 | 0.219 | 0.311 | 0.304 | 0.366 | 0.327 | 0.384 | 0.191 | 0.284 |
> |           | Weather| 0.255 | 0.276 | 0.256 | 0.277 | 0.264 | 0.285 | 0.293 | 0.322 | 0.265 | 0.286 | 0.295 | 0.308 | 0.270 | 0.287 | 0.261 | 0.282 |
>
>
>
>
> #### [W3] Evaluation is restricted to direct forecasting, limiting evidence of robustness across different training paradigms. **Additional experiments under an autoregressive setting** would be valuable to validate whether the proposed objective is broadly applicable across different forecasting architectures and training pipelines.
>
> **Response.** Thank you very much for your detailed suggestion. We address this concern through the following points.
> - Overall, DistDF is dedicated to **address the autocorrelation within the label sequence**. In the autoregressive setting, there is only one-step label in the label sequence in each update, so the **label autocorrelation does not seem to be present or significant**, which hampers the direct use of DistDF in this setting. Nevertheless, in the autoregressive setting, **different covariates in the label could have significant correlations**. It would be intuitive to investigate **whether DistDF can accommodate the covariate correlation and improve performance** in the autoregressive setting, to showcase the broad applicability of DistDF.
> - **Additional experiment.** We select TimeBridge and FredFormer as the forecasting models in this test due to their competitive performance. **We modify them to perform autoregression-based prediction**, and apply DistDF for training. Notably, here we use DistDF to model the covariate correlations, so the transport matrix size should be $D \times D$ instead of $T\times T$.
> - **Result analysis.** The results are presented below, where $^\dagger$ denotes models trained with DistDF. There are two key observations. (i) The overall performance under the autoregressive setting is generally inferior to that of the direct forecasting (DF) paradigm, which can be attributed to the accumulation of errors during iterative prediction, particularly for longer forecasting horizons. (ii) Models trained with DistDF ($^\dagger$) consistently outperform their counterparts, demonstrating that the utility of DistDF extends to the autoregressive regime.
> - **Revision.** We add the experimental results under autoregressive setting in Table 16.
>
> |           |        | TimeBridge | | TimeBridge$^\dagger$ | | Fredformer | | Fredformer$^\dagger$ | |
> |-----|----|---|--|---|---|----|-------|-------|-------|
> |           |        | MSE   | MAE   | MSE   | MAE   | MSE   | MAE   | MSE   | MAE   |
> | **ETTm1** | 96     | 0.405 | 0.402 | **0.395** | **0.391** | 0.391 | 0.396 | **0.386** | **0.390** |
> |         | 192    | 0.467 | 0.438 | **0.419** | **0.408** | 0.494 | 0.449 | **0.493** | **0.446** |
> |           | 336    | 0.518 | 0.467 | **0.460** | **0.437** | 0.572 | 0.500 | **0.579** | **0.486** |
> |           | 720    | 0.725 | 0.514 | **0.527** | **0.478** | 1.821 | 0.837 | **0.833** | **0.563** |
> |           | **Avg**| 0.528 | 0.455 | **0.450** | **0.428** | 0.820 | 0.546 | **0.573** | **0.471** |
> | **Weather**| 96    | 0.527 | 0.343 | **0.241** | **0.275** | 0.241 | 0.267 | **0.211** | **0.245** |
> |           | 192    | 1.165 | 0.494 | **0.303** | **0.320** | 0.306 | 0.318 | **0.274** | **0.292** |
> |           | 336    | 4.826 | 0.749 | **0.371** | **0.365** | 0.330 | 0.331 | **0.312** | **0.322** |
> |           | 720    | 9.363 | 1.374 | **0.461** | **0.421** | 0.433 | 0.406 | **0.407** | **0.380** |
> |           | **Avg**| 3.970 | 0.740 | **0.344** | **0.345** | 0.327 | 0.330 | **0.301** | **0.310** |

---

> ### Author Response · Authors · 2025-11-27
>
> #### [W4] In Table 1, it is **unclear which underlying model architectures DistDF is applied to**. Since DistDF is a learning objective rather than a new architecture, and the table compares against architectural baselines, the presentation may confuse readers regarding what is being evaluated. Clarifying the base model used for each dataset would improve readability. **Explicitly specifying the base architecture for each dataset (e.g., as done in Scaleformer, ICLR 2023)** would improve clarity and ensure a fair interpretation of the reported gains.
> **Response.** Thank you for this sincere suggestion. We address this concern by (i) clarifying the underlying model architectures in Table 1 and (ii) providing a new table following Scaleformer.
> - **Firstly, we clarify that in Table 1, DistDF employs the top-performing baseline on each dataset as the forecasting model** (highlighted in blue). This approach is consistent to pioneer works (FreDF [ICLR 2025], Time-o1 [NeurIPS 2025]). It allows us to evaluate whether DistDF is able to consistently enhance the performance of the best existing forecasting models across various datasets, even though the top-performing model may differ from one dataset to another.
>
> - **Secondly, we provide new results following the protocol of Scaleformer**, to evaluate how DistDF performs given specific forecasting models. We select forecasting models recently released: TimeBridge [ICML 2025], TQNet [ICML 2025], FredFormer [KDD 2024], iTransformer [ICLR 2024] and FreTS [NeurIPS 2023]. The results are available as follows, where DistDF can effectively improve different forecasting models.
>
> - **Revision.** We add a tablenote in Table 1: `DistDF employs the underlined baseline that performs best on each dataset as the forecast model.`, to address issue (i) above. Moreover, we add the table below in the revised manuscript (Table 15), to address issue (ii) above.
>
>
> | Dataset | TQNet (MSE) | TQNet (MAE) | TQNet$^\dagger$ (MSE) | TQNet$^\dagger$ (MAE) | TimeBridge (MSE) | TimeBridge (MAE) | TimeBridge$^\dagger$ (MSE) | TimeBridge$^\dagger$ (MAE) | Fredformer (MSE) | Fredformer (MAE) | Fredformer$^\dagger$ (MSE) | Fredformer$^\dagger$ (MAE) | iTransformer (MSE) | iTransformer (MAE) | iTransformer$^\dagger$ (MSE) | iTransformer$^\dagger$ (MAE) | FreTS (MSE) | FreTS (MAE) | FreTS$^\dagger$ (MSE) | FreTS$^\dagger$ (MAE) |
> | :--- | :---: | :---: | :---: | :---: | :---: | :---: | :---: | :---: | :---: | :---: | :---: | :---: | :---: | :---: | :---: | :---: | :---: | :---: | :---: | :---: |
> | **ETTh1** | 0.449 | 0.439 | 0.438 | 0.431 | 0.442 | 0.440 | 0.434 | 0.436 | 0.447 | 0.434 | 0.430 | 0.429 | 0.452 | 0.448 | 0.447 | 0.444 | 0.489 | 0.474 | 0.479 | 0.467 |
> | **ETTh2** | 0.375 | 0.400 | 0.371 | 0.399 | 0.377 | 0.403 | 0.373 | 0.398 | 0.377 | 0.402 | 0.367 | 0.393 | 0.386 | 0.407 | 0.379 | 0.405 | 0.524 | 0.496 | 0.466 | 0.467 |
> | **ETTm1** | 0.376 | 0.391 | 0.375 | 0.391 | 0.387 | 0.400 | 0.383 | 0.397 | 0.387 | 0.398 | 0.378 | 0.394 | 0.411 | 0.414 | 0.404 | 0.410 | 0.414 | 0.421 | 0.400 | 0.409 |
> | **ETTm2** | 0.277 | 0.321 | 0.272 | 0.318 | 0.281 | 0.326 | 0.280 | 0.323 | 0.280 | 0.324 | 0.277 | 0.321 | 0.295 | 0.336 | 0.289 | 0.330 | 0.316 | 0.365 | 0.306 | 0.358 |
> | **ECL** | 0.175 | 0.265 | 0.171 | 0.262 | 0.176 | 0.271 | 0.172 | 0.267 | 0.191 | 0.284 | 0.173 | 0.266 | 0.179 | 0.270 | 0.174 | 0.265 | 0.199 | 0.288 | 0.199 | 0.287 |
> | **Weather** | 0.246 | 0.270 | 0.244 | 0.269 | 0.252 | 0.277 | 0.248 | 0.275 | 0.261 | 0.282 | 0.255 | 0.277 | 0.269 | 0.289 | 0.258 | 0.280 | 0.249 | 0.293 | 0.245 | 0.288 |

---

> ### Author Response · Authors · 2025-11-27
>
> #### [Q1] Since the proposed objective must be combined with MSE for stable training, can the authors provide evidence that the improvement does not simply arise from a regularization effect? For example, **how does the discrepancy term alone behave, and how strongly does its reduction correlate with forecasting accuracy?**
>
> **Response.** Thank you for this insightful suggestion. As the aspect concerning “how strongly does its reduction correlate with forecasting accuracy” is addressed in our response to [W1], we focus here on the remaining aspect regarding the behavior of the discrepancy term when used in isolation. Our response is structured as follows:
>
> - **First, we add experiments to assess the effect of employing the discrepancy term as the sole objective.** Specifically, we evaluate three settings across four datasets: (1) direct forecasting using only MSE ($\alpha=0$), (2) the discrepancy term alone ($\alpha=1$), and (3) DistDF as a regularizer ($\alpha=\alpha^*$). The empirical results indicate that the discrepancy term alone does not lead to consistent improvements: performance degrades on the ECL and Weather datasets. In contrast, utilizing the discrepancy term as a regularizer alongside MSE consistently enhances direct forecasting performance, empirically validating our methodological choice, in line with established practices [1-2].
> - **Second, we elucidate the critical role that the MSE term plays in fully realizing the performance benefits of DistDF**. By Lemma 3.5, the Bures-Wasserstein discrepancy quantifies the divergence between the mean and covariance of the joint distributions of the forecast and label sequences. This term involves important distributional characteristics, but it discards elementwise correspondences between forecast and label sequences—information critical for forecasting tasks. While mini-batch stochastic gradient descent may partly compensate for this loss via sampling diversity, it cannot fully restore pairwise correspondence information. Therefore, the inclusion of MSE as an additional objective is needed to recover the pairwise correspondence, which is essential to train an accurate forecasting model.
>
> - **Revision.** We add limitation discussion in the conclusion section, explicitly stating that DistDF’s primary benefit is as a regularization term complementing standard MSE loss.
>
> | $\alpha$ | ETTm1 MSE | ETTm1 MAE | ETTm2 MSE | ETTm2 MAE | ECL MSE | ECL MAE | Weather MSE | Weather MAE |
> |--------|---------|---------|---------|---------|-------|-------|------------|--------|
> | $\alpha=0$      | 0.387 | 0.398 | 0.280 | 0.324 | 0.191 | 0.284 | 0.261 | 0.282 |
> | $\alpha=1$      | 0.404 | 0.408 | 0.283 | 0.326 | 0.239 | 0.326 | 0.268 | 0.290 |
> | $\alpha=\alpha^*$    | **0.380** | **0.395** | **0.277** | **0.322** | **0.175** | **0.267** | **0.255** | **0.278**
>
>
>
> [1] FreDF: Learning to Forecast in the Frequency Domain. ICLR 2025.
>
> [2] Time-o1: Time-Series Forecasting Needs Transformed Label Alignment. NeurIPS 2025.
>
>
> #### [Q2] The distinction between DistDF and existing learning-objective methods such as **Time-o1, FreDF, Koopman-based losses, and Soft-DTW** remains somewhat unclear. Can the authors more explicitly highlight the conceptual and practical differences, particularly regarding **theoretical guarantees** and **optimization behavior**?
>
> **Response.** Thank you for this insightful suggestion. We can compare these methods from three aspects:
> - Principle: What is the main idea of the method to enhance learning objective?
> - **Theoretical guarantees**: Are they theoretically guaranteed to fully remove the autocorrelation bias? (**Fully debiased**)
> - **Optimization behavior**: Are they learning objectives with analytical form? (**Analytical**) Are they model agnostic? (**Model agnostic**).
>
> | Method | Principle | Analytical | Fully debiased | Model agnostic |
> |-------|-------|-------|-------|-------|
> | Time-o1 | Label transformation | $\checkmark$ | $\times$ | $\checkmark$ |
> | FreDF | Label transformation | $\checkmark$ | $\times$ | $\checkmark$ |
> | Koopman | Dynamic system training acceleration | $\checkmark$ | $\times$ | $\times$ |
> | Soft-DTW | Shape alignment | $\checkmark$ | $\times$ | $\checkmark$ |
> | DistDF | Distribution alignment | $\checkmark$ | $\checkmark$ | $\checkmark$ |

---

> ### Author Response · Authors · 2025-11-27
>
> The comparison results are provided in the table above, and the detailed analysis is presented as follows.
> - **Time-o1 and FreDF** are label transformation methods, which transform the label sequence into latent components with reduced autocorrelation bias. They are model agnostic and have analytical forms. However, as discussed in the manuscript (section 3.1), they are not fully debiased since they can only generate marginal decorrelated components, not conditional decorrelated components, rendering residual bias.
> - **Koopman** is a dynamic system training acceleration method, which accelerates the training of dynamic system models by using FFT. It has an analytical form but is not model agnostic since it requires the underlying model to be an RNN-like dynamic system that performs recurrent inference for multi-step forecasts. Moreover, its research problem is not related to eliminating the autocorrelation bias.
> - **Soft-DTW** is a shape alignment method, which aligns the shape of the predicted and label sequences. It is model agnostic but does not have an analytical form: it requires solving a constrained optimization problem to obtain the value. Besides, although it considers label autocorrelation by treating forecasting as a shape alignment problem, no theoretical guarantees are provided to ensure that it eliminates the bias caused by label autocorrelation.
> - **DistDF** is a distribution alignment method, which aligns the conditional distribution of the predicted and label sequences. It has an analytical form and is model agnostic. Moreover, it has theoretical guarantees to eliminate the autocorrelation bias (see Theorem 3.4, alignment property).
>
> #### [Q3] The discussion of likelihood bias focuses primarily on MSE. **Do similar issues arise in probabilistic forecasting frameworks** using alternative objectives (e.g., quantile loss, CRPS)? If so, is DistDF compatible with or beneficial under such setups?
>
> **Response.** Thank you for your sincere comment. We agree that it is valuable to explore the utility of DistDF to enhance probabilistic forecasting frameworks.  Our response is structured as follows:
> - **Additional experiments.** We select D3U, the state-of-the-art probabilistic forecasting framework as the testbed. In the variant denoted as  D3U$^\dagger$, we regularize the learning objective of D3U with DistDF. The results are presented in the table below, where D3U$^\dagger$ exhibits consistent performance improvement, **validating the utility of DistDF in recent probabilistic forecasting frameworks.**
> - **Revision.** We add the table below in the revised manuscript (Table 17), to demonstrate the efficacy of DistDF to improve probabilistic forecasting frameworks.
>
> | Dataset   | Horizon | D3U (MSE) | D3U (MAE) | D3U (CRPS) | D3U (CRPS$_{sum}$) | D3U$^\dagger$ (MSE) | D3U$^\dagger$ (MAE) | D3U$^\dagger$ (CRPS) | D3U$^\dagger$  (CRPS$_{sum}$) |
> |-----------|---------|---------|---------|----------|------------------------|---------------------|---------------------|----------------------|-------------------------------|
> | **ETTm1** | 96      | 0.317   | 0.357   | 0.263    | 0.723                  | 0.316               | 0.357               | 0.265                | 0.720                         |
> |           | 192     | 0.361   | 0.383   | 0.285    | 0.749                  | 0.360               | 0.383               | 0.282                | 0.747                         |
> |           | 336     | 0.394   | 0.404   | 0.299    | 0.742                  | 0.390               | 0.402               | 0.298                | 0.731                         |
> |           | 720     | 0.460   | 0.437   | 0.325    | 0.892                  | 0.453               | 0.435               | 0.328                | 0.849                         |
> |           | **Avg** | 0.383   | 0.395   | 0.293    | 0.776                  | 0.380               | 0.394               | 0.293                | 0.762                         |
> | **Weather** | 96    | 0.176   | 0.240   | 0.174    | 0.179                  | 0.173               | 0.225               | 0.171                | 0.173                         |
> |    | 192     | 0.223   | 0.271   | 0.205    | 0.234                  | 0.217               | 0.265               | 0.198                | 0.210                         |
> |   | 336     | 0.279   | 0.309   | 0.233    | 0.269         | 0.278               | 0.310               | 0.233                | 0.260                         |
> |      | 720     | 0.359   | 0.361   | 0.273    | 0.419       | 0.353               | 0.360               | 0.269                | 0.378                         |
> |           | **Avg** | 0.259   | 0.295   | 0.221    | 0.275                  | 0.255               | 0.290               | 0.218                | 0.255                         |
>
> [3] Diffusion-based decoupled deterministic and uncertain framework for probabilistic multivariate time series forecasting. ICLR 2025.

---

> ### Author Response · Authors · 2025-11-27
>
> #### [Q4] How well does DistDF extend to **multivariate forecasting**, **probabilistic output formulations**, or **multi-scale architectures**? Providing results or analysis in these more general settings would help verify that the proposed approach is broadly applicable beyond the current scope.
>
> **Response.** We express sincere gratitude once again for the meticulous and helpful suggestions. We kindly note that: (i) In the current manuscript, the experiments are conducted under **multivariate forecasting** setting, i.e., involving multiple covariates in the label sequence. (ii) In the response to [Q3], the utility of DistDF for **probabilistic output formulations** has been investigated. Therefore, we focus here on the remaining aspect regarding **multi-scale architectures**.
>
> - **Additional experiments.** We select TimeMixer and SCINet, two competitive multi-scale architectures based on CNN and MLP respectively as the testbed. In the variants TimeMixer$^\dagger$ and SCINet$^\dagger$, we train the architectures via DistDF. The results are presented in the table below, where TimeMixer$^\dagger$ and SCINet$^\dagger$ consistently outperform TimeMixer and SCINet, respectively, **validating the utility of DistDF to improve the performance of multi-scale architectures.**
>
> - **Revision.** We add the table below in the revised manuscript (Table 18), to demonstrate the efficacy of DistDF to improve multiscale architectures.
> | Dataset   | TimeMixer (MSE) | TimeMixer (MAE) | TimeMixer$^\dagger$ (MSE) | TimeMixer$^\dagger$ (MAE) | SCINet (MSE) | SCINet (MAE) | SCINet$^\dagger$ (MSE) | SCINet$^\dagger$ (MAE) |
> |-----------|-----------------|-----------------|---------------------------|---------------------------|--------------|--------------|------------------------|------------------------|
> | **ETTm1**   | 0.422           | 0.423           | 0.401                     | 0.411                     | 0.418        | 0.416        | 0.389                  | 0.399                  |
> | **ETTh1**   | 0.501           | 0.476           | 0.456                     | 0.446                     | 0.467        | 0.450        | 0.459                  | 0.443                  |
> | **ECL**     | 0.176           | 0.273           | 0.172                     | 0.267                     | 0.173        | 0.273        | 0.169                  | 0.268                  |
> | **Weather** | 0.262           | 0.285           | 0.252                     | 0.280                     | 0.253        | 0.280        | 0.250                  | 0.278                  |

---

### Official Review · Reviewer_aY9s · 2025-11-01

**Soundness:** 3
**Presentation:** 2
**Contribution:** 3
**Rating:** 8
**Confidence:** 3

**Summary:**

The paper proposes a Wasserstein-based discrepancy measure for time series that captures label autocorrelation and demonstrates the benefits of using it for time series alignment compared to established methods. Several experiments were conducted to support this claim.

**Strengths:**

The presented Wasserstein discrepancy seems original and effective. The experiments seem comprehensive and well carried out.

**Weaknesses:**

I think the paper should discuss the assumption of Gaussian distributed data more. It seems absolutely necessary to derive the discrepancy measure and yet I suppose the benchmark datasets do not satisfy this property.

I consider this a mild weakness but the theory regarding the general Wasserstein metric is presented mostly for discrete measures. Given that a Gaussian data distribution is assumed, it could be discussed how the presented results for empirical measures relate to the original Gaussian data distribution.

Minor
-------
The Bures-Wasserstein discrepancy is spelled as “Bruce-Wasserstein” in Lemma 3.5. Also in this Lemma, the equality to the W_2 metric should be made clear.

**Questions:**

Perhaps I am missing something, but Table 1 seems confusing. DistDF, which is a discrepancy measure, is compared to other models. It should be pointed out which model was used with DistDF loss.

**Details Of Ethics Concerns:**

No ethics concerns.

---

> ### Author Response · Authors · 2025-11-27
>
> Thank you so much for your positive evaluation and appreciation of **our novelty and empirical studies**. Below are our responses to the specific query raised.
>
> ----
>
> #### [W1] I think the paper **should discuss the assumption of Gaussian distributed data more.** It seems absolutely necessary to derive the discrepancy measure and yet I suppose the benchmark datasets do not satisfy this property.
>
> **Response.** Thank you very much for your thoughtful comment. We acknowledge that the Gaussian distribution assumption may not precisely hold for real-world datasets. Nonetheless, we would like to clarify two points: (i) the adoption of the Gaussian assumption is a well-established and prevalent practice in the time-series forecasting literature, and (ii) the Bures-Wasserstein (BW) discrepancy remains instrumental for non-Gaussian data.
>
> - **First, we recognize that assuming Gaussianity underpins many standard methodologies and theoretical developments in time-series forecasting.** For instance, **data standardization**—a ubiquitous preprocessing step—assumes Gaussianity; several theoretical analyses (e.g., Theorem 3.1 in the pioneering work [1]) are grounded in this assumption. Our adoption of the Gaussian hypothesis is thus consistent with established works and **does not introduce stringent assumptions that are not used in established forecasting pipelines**.
>
> - Second, although the BW discrepancy is theoretically derived under the Gaussian assumption, **we note that it is effective to align non-Gaussian data.** Specifically, the BW discrepancy measures the difference of the **first and second moments (mean and covariance)**. While higher-order moments are necessary for a complete characterization of non-Gaussian distributions, the first and second moments nonetheless capture **the most salient distributional characteristics**. As such, BW discrepancy can substantially align distributions in practice, even in non-Gaussian settings.
>
> - **Revision.** **We have added a dedicated discussion regarding the limitations of the Gaussian assumption at the end of Appendix A.** We also highlight that BW discrepancy aligns distributions by matching first- and second-order moments, which remains effective to a large extent even when the data is not Gaussian.
>
>
> [1] Time-o1: Time-series forecasting needs transformed label alignment[C]//The Thirty-ninth Annual Conference on Neural Information Processing Systems. 2025.
>
> #### **[W2] I consider this a mild weakness but the theory regarding the general Wasserstein metric is presented mostly for discrete measures. Given that a Gaussian data distribution is assumed, it could be discussed how the presented results for empirical measures relate to the original Gaussian data distribution.**
>
> **Response.** Thank you for highlighting this important theoretical point. We agree that clarifying the relationship between the presented results for empirical measures and their Gaussian counterparts is necessary. We elaborate on this connection below:
>
> - **Firstly, we clarify the results for empirical measures.** The general Wasserstein discrepancy is introduced in Definition 3.2, which directly applies to empirical probability measures. In Section 3.1, we frame time-series forecasting as a distribution alignment problem, where ideally the $\mathcal{W}\_p$ discrepancy between conditional distributions, $\mathcal{W}\_p(\mathbb{P}\_{Y|X},\mathbb{P}\_{\hat{Y}|X})$, would be minimized to train the forecasting model. However, computing the discrepancy between the conditional distributions is intractable. To overcome this, Lemma 3.3 upper-bounds the conditional discrepancy by the tractable joint discrepancy $\mathcal{W}\_p(\mathbb{P}\_{X,Y}, \mathbb{P}\_{X,\hat{Y}})$. Nevertheless, using it as a learning objective is computationally expensive (due to the need for iterative optimization steps and gradient backpropagation through the iteration chains).
>
> - **Secondly, we clarify how the results above relate to Gaussian data distribution.** To further address the complexity above, in Lemma 3.5, we employ the Bures-Wasserstein discrepancy as a proxy to $\mathcal{W}\_p(\mathbb{P}\_{X,Y},\mathbb{P}\_{X,\hat{Y}})$, which has a closed-form expression and can be computed analytically. Moreover, it is a special form of $\mathcal{W}\_p(\mathbb{P}\_{X,Y},\mathbb{P}\_{X,\hat{Y}})$ under the Gaussian assumption, which is exactly the relationship between the results for empirical measures and the Gaussianity assumption.

---

> ### Author Response · Authors · 2025-11-27
>
> - One may wonder **how the Bures-Wasserstein discrepancy behaves when the Gaussianity assumption is not fully satisfied.** As addressed in the response to [W1], the Bures-Wasserstein discrepancy measures differences only in the first- and second-order moments—i.e., the mean and covariance. While these two moments fully characterize Gaussian distributions, non-Gaussian distributions additionally require higher-order moments for complete characterization. Nonetheless, the mean and covariance remain essential descriptors for any distribution. As a result, in cases where data deviate from strict Gaussianity, the Bures-Wasserstein discrepancy remains a valuable tool for distribution alignment by matching these fundamental moments.
>
> #### **[W3] The Bures-Wasserstein discrepancy is spelled as "Bruce-Wasserstein" in Lemma 3.5. Also in this Lemma, the equality to the $\mathcal{W}_2$ metric should be made clear.**
> **Response.** Thank you very much for your meticulous comment! We have revised these typographical errors in the revised manuscript and highlight the equality.
> - We have replaced `Bruce-Wasserstein` with `Bures-Wasserstein`.
> - We have highlighted the equality to the $\mathcal{W}_2$ metric in Lemma 3.5.
>
> #### [Q1] Perhaps I am missing something, but Table 1 seems confusing. DistDF, which is a discrepancy measure, is compared to other models. **It should be pointed out which model was used with DistDF loss.**
>
> **Response.** Thank you very much for your careful observation. We agree that it is necessary to point out which forecasting model was used with DistDF loss in Table 1.
>
> - Firstly, we clarify that in Table 1, DistDF employs the top-performing baseline (highlighted in blue) on each dataset as the forecasting model. This approach allows us to evaluate whether DistDF is able to consistently enhance the performance of the best existing forecasting models across various datasets, even though the top-performing model may differ from one dataset to another.
>
> - **Additional experiments.** We add experiments to evaluate how DistDF performs given specific forecasting models. We select forecasting models recently released: TimeBridge [ICML 2025], TQNet [ICML 2025], FredFormer [KDD 2024], iTransformer [ICLR 2024] and FreTS [NeurIPS 2023]. **The results are available as follows, where we explicitly point out which forecasting model is used.** Overall, DistDF can effectively improve different forecasting models.
>
> | Dataset | TQNet (MSE) | TQNet (MAE) | TQNet$^\dagger$ (MSE) | TQNet$^\dagger$ (MAE) | TimeBridge (MSE) | TimeBridge (MAE) | TimeBridge$^\dagger$ (MSE) | TimeBridge$^\dagger$ (MAE) | Fredformer (MSE) | Fredformer (MAE) | Fredformer$^\dagger$ (MSE) | Fredformer$^\dagger$ (MAE) | iTransformer (MSE) | iTransformer (MAE) | iTransformer$^\dagger$ (MSE) | iTransformer$^\dagger$ (MAE) | FreTS (MSE) | FreTS (MAE) | FreTS$^\dagger$ (MSE) | FreTS$^\dagger$ (MAE) |
> | :--- | :---: | :---: | :---: | :---: | :---: | :---: | :---: | :---: | :---: | :---: | :---: | :---: | :---: | :---: | :---: | :---: | :---: | :---: | :---: | :---: |
> | **ETTh1** | 0.449 | 0.439 | 0.438 | 0.431 | 0.442 | 0.440 | 0.434 | 0.436 | 0.447 | 0.434 | 0.430 | 0.429 | 0.452 | 0.448 | 0.447 | 0.444 | 0.489 | 0.474 | 0.479 | 0.467 |
> | **ETTh2** | 0.375 | 0.400 | 0.371 | 0.399 | 0.377 | 0.403 | 0.373 | 0.398 | 0.377 | 0.402 | 0.367 | 0.393 | 0.386 | 0.407 | 0.379 | 0.405 | 0.524 | 0.496 | 0.466 | 0.467 |
> | **ETTm1** | 0.376 | 0.391 | 0.375 | 0.391 | 0.387 | 0.400 | 0.383 | 0.397 | 0.387 | 0.398 | 0.378 | 0.394 | 0.411 | 0.414 | 0.404 | 0.410 | 0.414 | 0.421 | 0.400 | 0.409 |
> | **ETTm2** | 0.277 | 0.321 | 0.272 | 0.318 | 0.281 | 0.326 | 0.280 | 0.323 | 0.280 | 0.324 | 0.277 | 0.321 | 0.295 | 0.336 | 0.289 | 0.330 | 0.316 | 0.365 | 0.306 | 0.358 |
> | **ECL** | 0.175 | 0.265 | 0.171 | 0.262 | 0.176 | 0.271 | 0.172 | 0.267 | 0.191 | 0.284 | 0.173 | 0.266 | 0.179 | 0.270 | 0.174 | 0.265 | 0.199 | 0.288 | 0.199 | 0.287 |
> | **Weather** | 0.246 | 0.270 | 0.244 | 0.269 | 0.252 | 0.277 | 0.248 | 0.275 | 0.261 | 0.282 | 0.255 | 0.277 | 0.269 | 0.289 | 0.258 | 0.280 | 0.249 | 0.293 | 0.245 | 0.288 |
>
> - **Revision.** We add a tablenote in Table 1: `DistDF employs the underlined baseline that performs best on each dataset as the forecast model.` Moreover, we add the additional table above in Table 15 in the revised manuscript.

---

### Official Review · Reviewer_ZpGW · 2025-11-01

**Soundness:** 3
**Presentation:** 4
**Contribution:** 3
**Rating:** 6
**Confidence:** 4

**Summary:**

The paper addresses time series forecasting and proposes aligning the predictive conditional distribution with the true conditional distribution by minimizing the joint-distribution Wasserstein discrepancy. This approach mitigates the bias introduced by autocorrelation when using maximum log-likelihood objectives to train forecasting models.

**Strengths:**

* Good observation that both frequency and PCA components exhibit autocorrelation, which affects their learning bias; this provides a well-motivated basis for applying optimal transport theory.

* The incorporation of DistDF in existing frameworks is straightforward

* Comprehensive experiments; I appreciate the effort to compare with other distributional discrepancies and the application of DistDF to other approaches

**Weaknesses:**

* The central hypothesis of this work is that aligning conditional distributions is beneficial, and the authors provide theoretical justifications along with empirical evidence through forecasting error metrics. However, it is unclear whether the conditional distributions actually align for the best alpha values reported in Tables 5 and 6. In other words, the hypothesis is not directly evaluated in the experiments through distributional discrepancy, but rather indirectly through forecasting performance.

* Improvements wrt to existing SOTA methods look rather small; however, they are consistent across models and datasets

**Questions:**

Please address my first point in the weaknesses. If no distributional discrepancy needs to be shown in the experiments, please elaborate why.

---

> ### Author Response · Authors · 2025-11-27
>
> Thank you so much for your encouraging support and appreciation of **our observation, methodology, and empirical studies**. Below are our responses to the specific query raised.
>
> ---
>
> #### [W1,Q1] The central hypothesis of this work is that aligning conditional distributions is beneficial, and the authors provide theoretical justifications along with empirical evidence through forecasting error metrics. However, it is unclear whether the conditional distributions actually align for the best alpha values reported in **Tables 5 and 6**. In other words, the hypothesis is not directly evaluated in the experiments through distributional discrepancy, but rather indirectly through forecasting performance. If no **distributional discrepancy needs to be shown** in the experiments, please elaborate why.
>
> **Response.** Thank you for your insightful question. **We agree that it is essential to evaluate the distributional discrepancy**.
> - **Additional experiment.** We add experiments based on **Tables 2, 5, and 6** to evaluate distributional discrepancy. As stated in this paper, the conditional distribution discrepancy is intractable. Therefore, we report the joint distribution discrepancy, which is the upper bound of the conditional discrepancy. The metric is denoted as Disc in the tables below.
>
> **The joint discrepancy results as a supplement to Table 2.**
> | Model| Dataset  | **DistDF** | Time-o1 | FreDF  | Koopman | Dilate | LDTW  | Soft-DTW | DTW| DF |
> |-|-|:-:|:-:|:-:|:-:|:-:|:--:|:--:|:--:|:--:|
> | **TimeBridge** | ETTm1 | **0.230** | 0.231| 0.231  | 0.271| 0.231  | 0.231 | 0.238  | 0.237 | 0.232 |
> || ETTh1 | **0.326** | 0.331| 0.330  | 0.350| 0.352  | 0.352 | 0.340  | 0.344 | 0.332 |
> || ECL| **0.129** | 0.135| 0.137  | 0.139| 0.136  | 0.139 | 0.133  | 0.140 | 0.136 |
> || Weather  | **0.147** | 0.148| 0.149  | 0.157| 0.148  | 0.148 | 0.153  | 0.150 | 0.148 |
> | **Fredformer** | ETTm1 | **0.227** | 0.228| 0.231  | 0.232| 0.233  | 0.233 | 0.240  | 0.239 | 0.232 |
> || ETTh1 | **0.324** | 0.325| 0.333  | 0.349| 0.349  | 0.350 | 0.356  | 0.355 | 0.342 |
> || ECL| **0.130** | 0.133| 0.134  | 0.142| 0.140  | 0.144 | 0.153  | 0.151 | 0.143 |
> || Weather  | 0.148  | **0.148** | 0.149  | 0.150| 0.150  | 0.152 | 0.152  | 0.152 | 0.152 |
>
> **The joint discrepancy results as a supplement to Table 5.**
>
>
> | $\alpha$ | ETTh2 (MSE) | ETTh2 (MAE) | ETTh2 (Disc) | ECL (MSE) | ECL (MAE) | ECL (Disc) | Weather (MSE) | Weather (MAE) | Weather (Disc) |
> |:--:|:-:|:-:|:--:|:-:|:-:|:-:|:--:|:--:|:-:|
> | 0  | 0.377  | 0.403  | 0.292 | 0.176| 0.271| 0.136  | 0.252 | 0.277 | 0.148|
> | 0.001 | 0.378  | 0.402  | 0.292 | 0.172| 0.267| 0.130  | 0.250 | 0.276 | 0.148|
> | 0.002 | 0.377  | 0.402  | 0.291 | 0.173| 0.267| **0.130**| 0.250 | 0.276 | 0.148|
> | 0.005 | 0.376  | 0.401  | 0.291 | **0.172**| **0.267**| 0.130  | 0.250 | **0.276**| 0.148|
> | 0.01  | 0.376  | 0.400  | 0.291 | 0.172| 0.267| 0.130  | **0.249**| 0.276 | **0.146**  |
> | 0.02  | 0.376  | 0.400  | 0.291 | 0.174| 0.269| 0.133  | 0.249 | 0.276 | 0.147|
> | 0.05  | **0.375** | 0.399  | **0.290**| 0.174| 0.268| 0.132  | 0.251 | 0.278 | 0.147|
> | 0.1| 0.375  | **0.399** | 0.291 | 0.174| 0.269| 0.132  | 0.254 | 0.280 | 0.148|
> | 0.2| 0.376  | 0.399  | 0.291 | 0.177| 0.270| 0.134  | 0.254 | 0.280 | 0.148|
> | 0.5| 0.378  | 0.400  | 0.294 | 0.186| 0.277| 0.140  | 0.256 | 0.281 | 0.149|
> | 1  | 0.381  | 0.402  | 0.296 | 0.197| 0.282| 0.147  | 0.260 | 0.283 | 0.150|
>
>
> **The joint discrepancy results as a supplement to Table 6**
>
> | $\alpha$ | ETTh2 (MSE) | ETTh2 (MAE) | ETTh2 (Disc) | ECL (MSE) | ECL (MAE) | ECL (Disc) | Weather (MSE) | Weather (MAE) | Weather (Disc) |
> |:--:|:-:|:-:|:--:|:-:|:-:|:-:|:--:|:--:|:-:|
> | 0  | 0.377  | 0.402  | 0.293 | 0.191| 0.284| 0.143  | 0.261 | 0.282 | 0.152|
> | 0.001 | 0.371  | 0.397  | 0.287 | 0.175| 0.268| 0.132  | **0.255**| **0.278**| **0.148**  |
> | 0.002 | 0.372  | 0.398  | 0.289 | **0.175**| **0.267**| **0.131**| 0.256| 0.278 | 0.149|
> | 0.005 | 0.372  | 0.398  | 0.288 | 0.182| 0.275| 0.137  | 0.256 | 0.279 | 0.149|
> | 0.01  | 0.370  | 0.397  | 0.285 | 0.183| 0.275| 0.137  | 0.257 | 0.279 | 0.150|
> | 0.02  | **0.369** | **0.395** | 0.286 | 0.182| 0.275| 0.136  | 0.258 | 0.280 | 0.149|
> | 0.05  | 0.370  | 0.396  | **0.285**| 0.187| 0.279| 0.141  | 0.259 | 0.281 | 0.150|
> | 0.1| 0.371  | 0.397  | 0.288 | 0.196| 0.287| 0.148  | 0.261 | 0.283 | 0.151|
> | 0.2| 0.372  | 0.398  | 0.290 | 0.209| 0.298| 0.158  | 0.263 | 0.285 | 0.152|
> | 0.5| 0.376  | 0.399  | 0.292 | 0.230| 0.317| 0.171  | 0.266 | 0.287 | 0.153|
> | 1  | 0.386  | 0.406  | 0.299 | 0.239| 0.326| 0.177  | 0.268 | 0.290 | 0.154|

---

> ### Author Response · Authors · 2025-11-27
>
> - **Result analysis.** There are two primary observations. (i) DistDF consistently achieves the lowest Disc values across various learning objectives, providing empirical evidence for the effectiveness of DistDF in aligning distributions and the associated benefits for forecasting accuracy. (ii) The values of $\alpha$ that minimize MSE and Disc exhibit strong concordance across different datasets, further supporting the claim that distributional alignment is conducive to improved forecasting performance.
> - **Revision.** **We have added the distributional discrepancy results as a supplement in Appendix D.8 (Table 12-14).**
>
> #### **[W2] Improvements wrt to existing SOTA methods look rather small; however, they are consistent across models and datasets.**
>
> **Response.** Thank you very much for your kind comment and recognition of the improvement consistency.
> - Firstly, we agree that compared to SOTA learning objective, typically Time-o1, the performance improvement achieved by DistDF is consistent yet moderate. However, we note that **with respect to the prevalent direct forecasting (DF) approach using MSE as objective, DistDF yields very substantial gains**, which provides empirical validation on the limitation of MSE and the benefit of DistDF to enhance forecasting performance.
> - Secondly, as stated in your strength 1, DistDF contributes beyond empirical improvements in two key aspects: (i) it **identifies an inherent theoretical limitation** of the Time-o1 learning objective; and (ii) it **reformulates time-series forecasting as a distribution alignment challenge**, underscoring the importance of distribution alignment. This establishes conceptual links to research areas based on distribution alignment such as transfer learning, which brings new insights to the time-series forecasting field.
> - Finally, to further substantiate the consistency of DistDF's improvements, we incorporate an additional learning objective (LDTW) as a supplementary baseline. The results are available as follows.
>
>
> | Model | Dataset  | **DistDF** (MSE) | **DistDF** (MAE) | Time-o1 (MSE) | Time-o1 (MAE) | FreDF (MSE) | FreDF (MAE) | Koopman (MSE) | Koopman (MAE) | Dilate (MSE) | Dilate (MAE) | LDTW (MSE) | LDTW (MAE) | Soft-DTW (MSE) | Soft-DTW (MAE) | DTW (MSE) | DTW (MAE) | DF (MSE) | DF (MAE) |
> |:--:|:--:|:--:|:--:|:--:|:--:|:-:|:-:|:-:|:-:|:--:|:--:|:-:|:-:|:-:|:-:|:-:|:-:|:-:|:-:|
> | **TimeBridge** | ETTm1 | **0.383**| **0.397**| 0.383 | 0.397 | 0.386| 0.398| 0.460  | 0.438  | 0.387 | 0.400 | 0.387 | 0.400 | 0.395 | 0.402 | 0.394| 0.401| 0.387  | 0.400  |
> | | ETTh1 | **0.434**| **0.436**| 0.439 | 0.438 | 0.439| 0.436| 0.459  | 0.449  | 0.464 | 0.452 | 0.464 | 0.452 | 0.452 | 0.445 | 0.455| 0.446| 0.442  | 0.440  |
> | | ECL| **0.172**| **0.267**| 0.175 | 0.268 | 0.175| 0.267| 0.182  | 0.277  | 0.176 | 0.271 | 0.182 | 0.275 | 0.173 | 0.268 | 0.178| 0.273| 0.176  | 0.271  |
> | | Weather  | **0.248**| **0.275**| 0.250 | 0.275 | 0.254| 0.276| 0.269  | 0.293  | 0.252 | 0.277 | 0.252 | 0.277 | 0.260 | 0.280 | 0.253| 0.276| 0.252  | 0.277  |
> | **Fredformer** | ETTm1 | **0.378**| 0.394 | 0.379 | **0.393**| 0.384| 0.394| 0.389  | 0.400  | 0.389 | 0.400 | 0.389 | 0.400 | 0.397 | 0.402 | 0.396| 0.402| 0.387  | 0.398  |
> | | ETTh1 | **0.430**| **0.429**| 0.431 | 0.429 | 0.438| 0.434| 0.452  | 0.443  | 0.453 | 0.442 | 0.453 | 0.442 | 0.460 | 0.449 | 0.460| 0.449| 0.447  | 0.434  |
> | | ECL| **0.173**| **0.266**| 0.178 | 0.270 | 0.179| 0.272| 0.190  | 0.282  | 0.187 | 0.280 | 0.193 | 0.284 | 0.206 | 0.298 | 0.202| 0.294| 0.191  | 0.284  |
> | | Weather  | **0.255**| 0.277 | 0.255 | **0.276**| 0.256| 0.277| 0.257  | 0.279  | 0.258 | 0.280 | 0.260 | 0.281 | 0.261 | 0.280 | 0.260| 0.280| 0.261  | 0.282  |

---

### Author Response · Authors · 2025-12-02
**Author Final Remarks**

Dear AC and all reviewers,

We sincerely appreciate your great efforts in evaluating our paper despite your busy schedules.  We are encouraged that 3 out of 4 reviewers are currently in the positive side, with all scores 8, 6, 6, and 4, recognizing our paper
- `provides a well-motivated basis` (Reviewer ZpGW),
- `seems original and effective`, with experiments `comprehensive and well carried out` (Reviewer aY9s),
- has `strong and clearly articulated motivation` and `solid theoretical foundation` (Reviewer c2pc),
- `well-motivated and offers an extremely thorough explanation` (Reviewer 6qSb).

Meanwhile, Reviewer c2pc gave an initial rating 4 with no further response. Due to the updated policy of ICLR-26, we understand Reviewer c2pc cannot provide us with further discussions. For facilitate checking, we summarize that the c2pc’s initial concerns are mainly on (1) Optimization dynamics and correlation with performance, (2) Presentation of experiment results, and (3) Performance on different scenarios, with our responses detailed as follows.

> Additional empirical analysis of its **optimization dynamics** and **correlation** with performance would strengthen the claims.
- We add experiments on the **optimization dynamics** of the proposed loss function (Figure 8).
- We quantify the **correlation** between the performance improvement trajectory and the optimization dynamics of the proposed loss function. It reveals a strong and consistent correlation, demonstrating that forecast accuracy consistently improves as the loss function is minimized

> Explicitly **specifying the base architecture** for each dataset (e.g., as done in **Scaleformer**, ICLR 2023) would improve clarity and ensure a fair interpretation of the reported gains.
- We add experiments **specifying 5 different base architectures** (Table 15), following the format of **Scaleformer**.
- We add a tablenote in Table 1 to **specify the base architectures** on different datasets.

> How well does DistDF extend to **multivariate forecasting**, **probabilistic output formulations**, or **multi-scale architectures**? Additional experiments under an **autoregressive setting** would be valuable.
- We add experiments to apply DistDF to **probabilistic output formulations**,  **multi-scale architectures** and **autoregressive settings**. In all three scenarios, DistDF consistently improved the performance of the base architectures.
- Regarding multivariate forecasting, **we respectfully clarify that the main results presented in the manuscript were already conducted under a multivariate setting**, confirming the method's effectiveness in this scenario.

We are confident that our responses can thoroughly address Reviewer c2pc's concerns.  Considering Reviewer c2pc doesn’t response to us after we posting the rebuttal, we respectfully ask you to consider this context when making your final recommendation.

Thank you in advance,

Authors of #8696

---

### Meta-Review · Area_Chair_2wmx · 2026-01-10

**Summary:**

I'd recommend acceptance. The paper makes a clean objective-level point—MSE/NLL proxies can be systematically off in multi-step forecasting when future labels are conditionally correlated—and offers a simple, model-agnostic fix by regularizing with a joint-distribution Wasserstein/Bures discrepancy. The reviewers who were positive liked the framing and the plug-and-play nature; the pushback was mainly about whether the gains are meaningful against strong recent baselines, whether the “conditional alignment” story was actually evidenced (vs just better MAE/MSE), and whether the discrepancy term is principled or simply an auxiliary regularizer that needs MSE to behave.

**Reviewer Concerns:**

I think the rebuttal did the right things for the main technical asks: it adds an explicit discrepancy metric (even if joint, as a surrogate), shows optimization dynamics and correlation with forecast error, and clarifies comparison protocol by training the same backbone with/without DistDF (including stronger recent models), plus it cleans up the confusing “DistDF-as-a-model” presentation and discusses the Gaussian assumption more honestly. What’s still hanging is mostly judgment-call territory: the improvements can still look incremental, the conditional-alignment claim is still supported indirectly (via a joint upper bound), and the method’s best story is still “good regularizer paired with MSE,” which some reviewers may see as less fundamental than the framing implies.

**Reviewer Scores:**

None of the reviewers gets to participant in discussions. I’d expect modest upward movement but not a full consensus: ZpGW’s main blocker was missing alignment evidence, so I’d guess 6->7; aY9s was already positive and mostly wanted clarity, so remaining 8; 6qSb’s baseline concern is largely addressed by same-backbone comparisons, so I'd expect a raise from 6 to 7; c2pc is the toughest: rebuttal answers many of their requests, but the remaining “proxy + regularizer” skepticism likely keeps them cautious, my guess is that they'd still raise their score to 4->5 in the best case.

---

### Decision · Program_Chairs · 2026-01-26

Accept (Poster)